# Tetris-inspired detector with neural network for radiation mapping

Ryotaro Okabe [1,2,8] ✉, Shangjie Xue[1,3,4,8], Jayson R. Vavrek[5,8], Jiankai Yu [3], Ryan Pavlovsky[5], Victor Negut[5], Brian J. Quiter[5], Joshua W. Cates[5], Tongtong Liu[1,6], Benoit Forget[3], Stefanie Jegelka [4], Gordon Kohse[7], Lin-wen Hu [7] ✉ & Mingda Li [1,3] ✉

Radiation mapping has attracted widespread research attention and increased public concerns on environmental monitoring. Regarding materials and their configurations, radiation detectors have been developed to identify the position and strength of the radioactive sources. However, due to the complex mechanisms of radiation-matter interaction and data limitation, high-performance and low-cost radiation mapping is still challenging. Here, we present a radiation mapping framework using Tetris-inspired detector pixels. Applying inter-pixel padding for enhancing contrast between pixels and neural networks trained with Monte Carlo (MC) simulation data, a detector with as few as four pixels can achieve high-resolution directional prediction. A moving detector with Maximum a Posteriori (MAP) further achieved radiation position localization. Field testing with a simple detector has verified the capability of the MAP method for source localization. Our framework offers an avenue for high-quality radiation mapping with simple detector configurations and is anticipated to be deployed for real-world radiation detection.

Since the Fukushima nuclear accident in 2011 till the recent risk at Zaporizhzhia nuclear power plant, there is an increasing global need calling for improved radiation detection technology, aiming to achieve high-performance radiation detection mapping with minimum impact on detectors and reduced cost. Due to the simultaneous presence of multiple radiation-interaction mechanisms, radiation detection for ionizing radiation is considerably harder than visible light. The large penetration depth of radiation, such as hard X-ray, γ-ray, and neutron, reduces the angular sensitivity of detectors and limits the majority of radiation detection efforts to focus on counting or spectra acquisition rather than their directional information. The challenge of acquiring directional radiation information further triggers additional difficulties in performing source localization, that to determine the position

distributions of radiation sources[1,2]. In recent years, radiation localization has attracted increased interest with applications such as autonomous nuclear site inspection. Several prototypes of system configurations have been proposed including unmanned ground[2-4], aerial[3,5-7] and underwater vehicles[8,9]. Despite the remarkable progress, the information extraction process of the radioactive environment is still seeking active developments.

In past decades, several approaches have been proposed for directional radiation detection. One approach is the High-Efficiency Multimode Imager (HEMI), which can be used to detect and locate γ-ray and X-ray radioactive sources[10-13]. A typical HEMI consists of two layers of CdZnTe (CZT) detectors; the first layer has a randomly arranged aperture for coded aperture imaging, and the second layer is

[1]Quantum Measurement Group, Massachusetts Institute of Technology, Cambridge, MA 02139, USA. [2]Department of Chemistry, Massachusetts Institute of Technology, Cambridge, MA 02139, USA. [3]Department of Nuclear Science and Engineering, Massachusetts Institute of Technology, Cambridge, MA 02139, USA. [4]Department of Electrical Engineering and Computer Science, Massachusetts Institute of Technology, Cambridge, MA 02139, USA. [5]Applied Nuclear Physics Program, Lawrence Berkeley National Laboratory, Berkeley, CA 94720, USA. [6]Department of Physics, Massachusetts Institute of Technology, Cambridge, MA 02139, USA. [7]Nuclear Reactor Laboratory, Massachusetts Institute of Technology, Cambridge, MA 02139, USA. [8]These authors contributed equally: Ryotaro Okabe, Shangjie Xue, Jayson R. Vavrek. ✉e-mail: rokabe@mit.edu; lwhu@mit.edu; mingda@mit.edu

the conventional co-planar detector grid. This system requires the incident beam to only come from a limited solid angle range to make sure the beam passes through the aperture of the first layer to interact with the second layer. The traditional reconstruction algorithm requires all the incident beams to come within the field of view. The accuracy will be affected if the radiation is incident from another direction (especially for near-field radiation). Besides, this system can only conditionally detect multiple sources, usually when the sources come from different isotopes and can be distinguished by energy. In this scenario, the detection with multiple sources can be reduced to single-source detection by only considering the count of events within an energy range. However, in real-world applications, different sources are not necessarily distinguishable in the energy spectrum. Besides HEMI, another approach for directional radiation detection is realized by using single pad detectors separated by padding material[14] i.e., self-shielded method. Radiation sources from different directions and distances can result in different intensity distribution patterns over detector arrays. Because of the inaccuracy of the model caused by misalignment and manufacturing errors of detector and shielding material, it is challenging to extract information from detector data via a traditional method such as non-linear fitting. Also, the traditional method is often most efficient in single source with reduced efficiency in multiple sources. As for radiation localization and mapping, inspired by the widespread interest in Simultaneous Localization and Mapping (SLAM)[15] techniques, several works using non-directional detectors[6,7] or HEMI[13] for radiation source localization and mapping have been presented. Recently, active masked designs have been developed, contrasting with traditional coded masks. In these designs, multiple detector segments shield each other, creating an anisotropically sensitive and omnidirectional field of view. For example, recent projects developed the neutron-gamma localization and mapping platform (NG-LAMP) system with a $2 \times 2$ array of CLLBC ($Cs_2LiLa(Br,Cl)_6$:Ce) detectors[16]; the MiniPRISM system[17,18], which uses a partially-populated $6 \times 6 \times 4$ array of CZT detectors; and an advanced neutron- and gamma-sensitive imager using a partially-populated $6 \times 6 \times 4$ array of CLLBC detectors[19]. There have been similar kinds of high-angular-contrast designs with additional high-density passive elements[20,21], or more traditional Compton cameras for in-field use[22–24].

In this work, we propose a radiation detection framework using a minimal number of detectors, combining Tetris-shaped detector with inter-pixel paddings, along with a deep-neural-network-based detector reading analysis. Figure 1 shows the overview of our framework. We demonstrate that detectors comprised of as few as four pixels, augmented by the inter-pixel padding material to intentionally increase contrast, could extract directional information with high accuracy. Moreover, we show that the shapes of the detectors do not have to be limited to a square grid. Inspired by the famous video game of Tetris, we demonstrate that other shapes from the Tetrominoes family, in which the geometric shapes are composed of four squares, can have potentially higher resolution (Fig. 1a). For each shape of the detector, we generate the data of the detector's input from radiation sources using Monte Carlo (MC) simulation (Fig. 1b). Figure 1c shows the machine learning model we trained to predict the direction of radiation sources. Using the filter layer and the deep U-net convolutional neural networks, we establish the model to predict the radiation source direction from the detected signal. As Fig. 1d illustrates, we compare the ground-truth label of the radiation source direction (blue) with the predicted direction (brown). By using Wasserstein distance as the loss function (see "Methods" for details), the model can achieve high accuracy of direction estimation. As an application of the directional detector, additional Maximum a Posteriori (MAP) has been implemented to a moving detector so that we can further estimate the spatial position of the radiation sources in both simulations and real-world experiments. Throughout this work, we limit the discussion to 2D since it is sufficient in many realistic scenario and leave the 3D

discussion for future studies. It's important to note that the radiation source extends beyond just gamma radiation. Each type of radiation necessitates the development of specialized detectors, as the properties of penetration, scattering, and detection mechanisms vary significantly among different radiation sources. For neutron localization tasks, significant advancements in neutron localization have been made with the high-efficiency, fast, and high-resolution thermal neutron imaging detectors[19,25–27]. Our work focuses on mapping $\gamma$ radiation. Localization of other radiation source types is expected to share similar principles, but is dedicated to future works.

## Results

### Directional prediction with static detectors

First, we train the machine learning model so that the static detectors can detect the direction from which the radiation comes from. We use OpenMC[28] package for MC simulation of the radiation detector receiving the signal from an external radiation source (more details in "Methods" Section). We assume that the detector pixels are composed of CZT detectors with pixel size $1 \text{ cm} \times 1 \text{ cm}$, slightly larger than the current crystals but still much smaller than the 5 meters of source-detector distance. The inter-pixel padding material is chosen to be 1 mm-thick lead empirically, which is thick enough to create contrast and with quite a low photon absorption in the $\gamma$-ray range. Throughout this study, we assume that the incident beam energy is $\gamma$-ray of 0.5 MeV, which is the realistic energy from pair production and comparable to many energy $\gamma$-decay energy levels. Given the energy resolution from CZT detector, radiations with other energies are also expected to be resolvable, even though here we only focus on the directional mapping where only counting matters. More details on the data preparation, normalization, neural network architectures and training procedures are shown in Methods Section. We evaluate the prediction accuracy of detectors which comprise with four detector configurations: $2 \times 2$ square grid, Tetrominos of S-, J-, T-shapes. The I-shaped Tetris detector array is not presented since it does not show performance good enough for directional mapping. The main results of the predicted radiation direction for the four Tetris-inspired detectors are illustrated in Fig. 2 and summarized in Table 1. While the S-shape detector worked with the smallest prediction followed by $2 \times 2$ square, J- and T-shapes, all of the four types of detector could work enough to know the direction of the radiation source with about 1-deg(°) accuracy.

Figure 2 shows detector readouts and typical angular distributions predicted by neural networks in polar plots. The blue and brown colors represent the ground truth from MC and the neural network prediction, respectively. Figure 2a, b are the $2 \times 2$ square-grid detector, showing a predictive power of the radiation with the largest and smallest prediction errors. The performance of other Tetris-inspired detector shapes, including S-, L- and T-shapes, are shown in Fig. 2c–h. By comparing different Tetris shapes, we can see that there is a generic trend that S-shaped Tetris can show the best performance while T-shaped Tetris is the least accurate. This can be intuitively understood from a symmetry analysis. For instance, for incident radiation from the "north" with $\theta = 0°$, the left and right pixels and padding materials in the T-shape receive identical signals from radiation sources, which reduces the number of effective pixels and padding materials. On the other hand, the S-shaped detector possesses pixels each of which receive non-equivalent signals from radiation sources of any direction, presenting lower prediction accuracy than the square detector. Although the square Tetris in Fig. 2a, b has higher symmetry than others, it also contains four pieces of inter-pixel padding materials, in contrast to other cases with three pieces. Such analysis may also apply to the I-shaped detector array, given its high symmetry and less effective pixels. Figure S4 provides the detailed analysis of the prediction accuracy with respect to the radiation source directions. In Supplementary Note 3, we present further analysis of directional

prediction with the Tetris-inspired detectors. Figure S5 shows the effect of the radiation source energy, arguing the importance of optimizing the radiation source energy for generating training dataset for practical application of the detector. Also, we proposed our proposed model architecture with two filter layers is effective in diverse scenarios by comparing it with the benchmark model of a single filter layer. Figure S6 and Table S3 show that the single-filter layer model has the capability to predict the directions of two radiation sources simultaneously. However, the two-filter layer model offers could accommodate diverse scenarios, as explained in Table S4. Furthermore, we surveyed the robustness of our directional prediction against the background noise effect. Figure S7 presented that the S-shaped detector performs the best if there is no noisy background, while the 2 × 2 detector had the highest robustness against the added Gaussian noise.

## Positional prediction with moving detectors and maximum a posteriori (MAP) estimation

In real-world applications of radiation mapping, it would be highly desirable to go beyond the directional information and also determine the position of the radiation source. Here, a method based on Maximum A Posteriori (MAP) estimation is proposed in order to generate the guessed distribution of radiations through the motion of detectors. The workflow is as follows: first, the detector readout is simulated by MC given the detector's initial position and orientation, just as the case for the static detector. Second, the detector begins to move in a circular motion. The schematics are shown in Fig. 3a. It is worthwhile mentioning that the particular detector face that aligns with the detector's moving direction does not matter much since the detector facing any direction is already a valid directional detector that is sensitive to radiations coming from all directions. In other words, even if the detector is rotated intentionally or accidentally during the circular motion, the final results are still robust (Figs. S15–S20 and Supplementary Movies 7–10 in Supplementary Note 6). Third, at each instantaneous timestamp during the detector motion, the predicted source direction is calculated based on the deep U-net model, just like the static detector case. Finally, the radiation source location is estimated via MAP based on the series of neural-network-inferred detector direction data at different detector positions. In an ideal case for one single isotropic radiation source, as few as two detector spatial

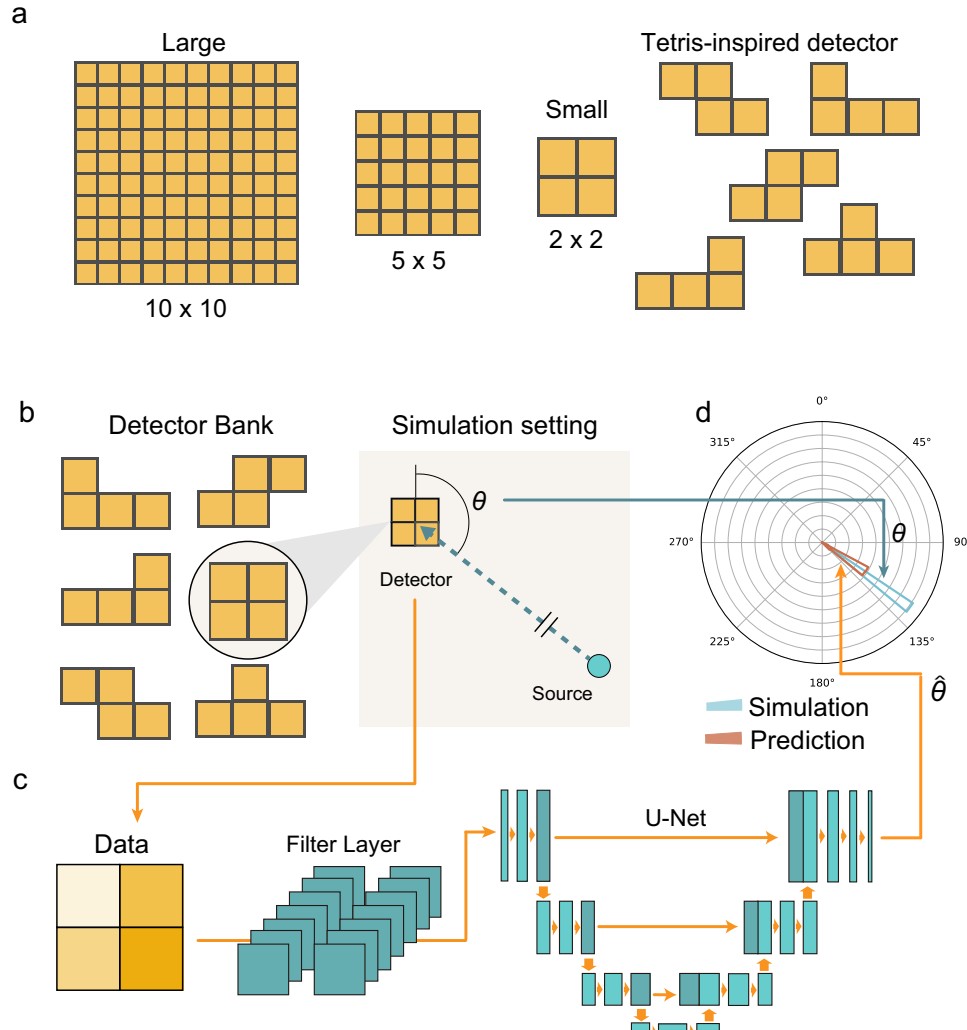

**Fig. 1 | Overview of Tetris-inspired radiation mapping with neural networks.** **a** The geometrical setting of the radiation detectors. Instead of using a detector with a large square grid, here we use small 2 × 2 square and other Tetromino shapes. Padding material is added between each pixel to increase contrast. **b**–**d** The workflow for learning the radiation directional information with Tetris-shaped detector and machine learning. **b** Monte Carlo simulation is performed to generate the detector readings for various source directions. **c** The detector's readouts are embedded to a matrix of filter layers for better distinguishing far-field and near-field scenarios. The embedded data then goes through a deep U-net. **d** The predicted direction of radiation sources from the U-net (brown) with predicted angular $\hat{\theta}$ is compared to the ground-truth Monte Carlo simulations (blue) with true angular data $\theta$. The prediction loss is calculated by comparing the pairs $(\theta, \hat{\theta})$.

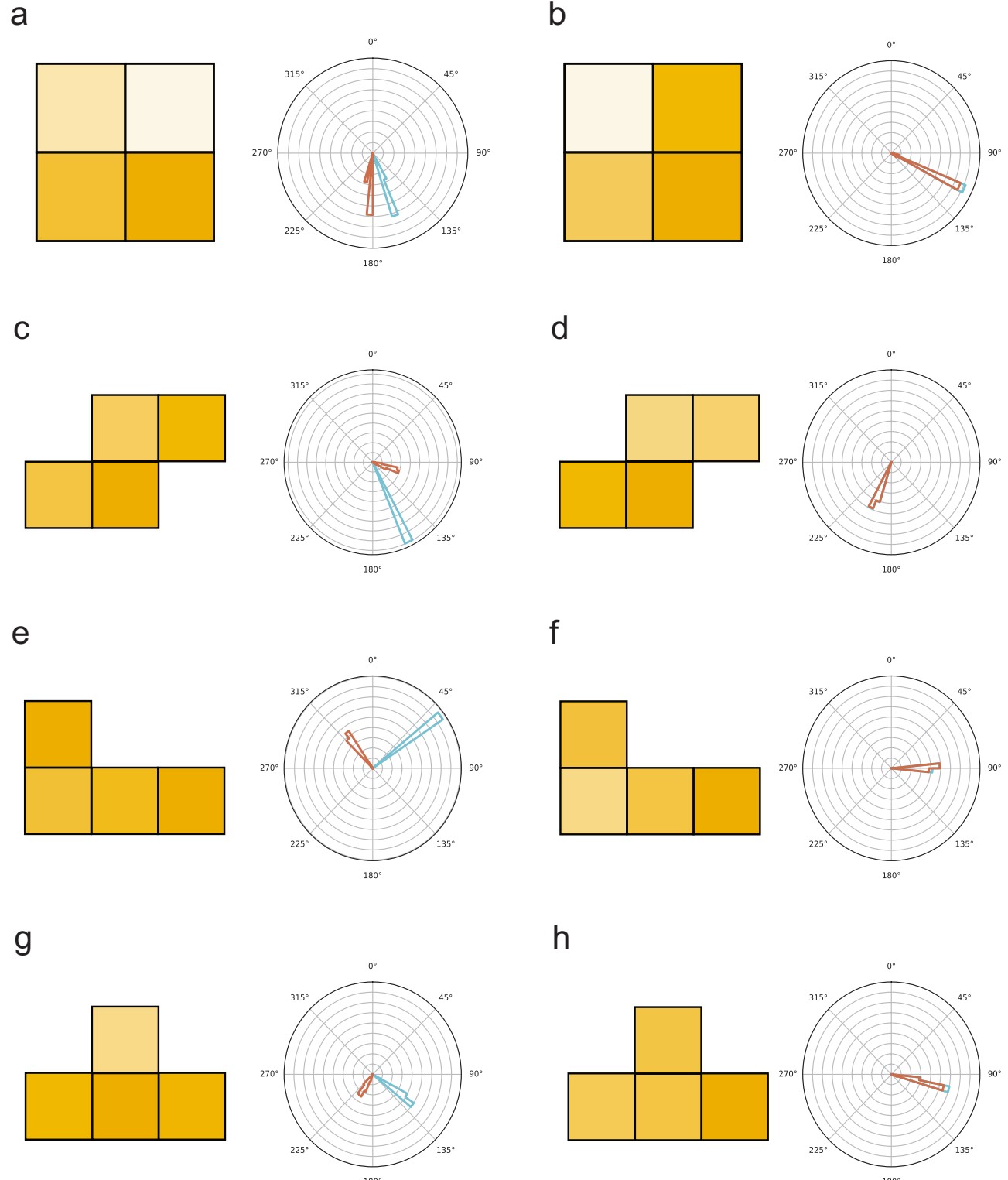

**Fig. 2 | Directional radiation detection with prediction with static Tetris detectors.** Selected outcomes from source direction predictions with detectors of simple configurations are displayed. Each figure includes the signal input employed for the input from the detector and the polar coordinates showing the directional predictions. The blue and brown curves represent the ground truth and prediction, respectively. We show results using the detector configurations of (**a**, **b**) a 2 × 2 square, (**c**, **d**) S-shape, (**e**, **f**) J-shape, and (**g**, **h**) T-shape. For each detector type, we display (**a**, **c**, **e**, **g**) the scenarios demonstrating the largest loss for each detector shape in test data, highlighting challenging prediction situations. **b**, **d**, **f**, **h** The scenarios with the smallest loss in test data, showcasing relatively successful prediction cases.

**Table 1 | Ablation on neural network backbones with different scenarios 2 × 2 square grid and three types of Tetrominos, evaluated by the Wasserstein distance of angular distributions**

|                | 2 × 2 | S-shape | J-shape | T-shape |
|----------------|-------|---------|---------|---------|
| Largest error  | **4.886** | 8.312 | 16.206 | 15.620 |
| Smallest error | 0.228 | **0.014** | 0.060 | 0.081 |
| Average error  | 1.114 | **0.949** | 1.563 | 1.775 |

The table shows the largest, smallest, and average prediction losses, respectively. The best value in each row is noted in bold.

positions are just enough to locate the source position (as the intersection of the two rays along the directions in the two detector positions) and the circular motion and MAP are implemented for more complicated radiation profile mapping. To improve the performance, we set a threshold for visualizing the radiation map. Our radiation maps are normalized by the highest probability of the presence of the radiation sources in the area of interest. We set the threshold as 0.3 and made every value on the area of interest that is lower than this threshold zero. This procedure enables us to visualize the mapping result clearly.

Figure 3b–j shows the dynamical process of radiation mapping and position determination using the S-shaped detector. The detector geometry is the same as before, and the radius of motion is chosen randomly distributed from 0.5 m to 5 m. Figure 3b–d shows the inferred radiation mappings at three different timestamps $t = 10, 30, 60$ s at the beginning, half-circle, and close to the end of the circular motion. The ground-truth location of the radiation source is shown as the black cross in all three figures. At the early $t = 10$ s, there is not sufficient information for MAP to estimate the radiation position, and the estimation (red lines in Fig. 3b) has a ray shape that acts more like directional mapping. After 30 s, the MAP estimation is improved, though the estimated radiation is located at a broader spatial area rather than the ground truth. Finally, the detector could complete the mapping process with sufficient accuracy to point out the position of the radiation source (Fig. 3d). The detector's readouts and the predicted directions at each timestamp ($t = 10, 20, 30, 40, 50,$ and 60 s) are illustrated as Fig. 3e–j. The detailed results with other Tetris-inspired detectors are shown in Supplementary Note 4. Figures S9–S11 present the moving detector and radiation mapping using the detectors of 2 × 2 square, J-shape and T-shape, respectively. Supplementary Movies 2–4 visualize these mapping processes at each timestamp. The actual and predicted relative angle are plotted throughout the observed time in Fig. S12.

When performing a realistic radiation mapping, the area of interest may contain multiple radiation sources, which increases the level of difficulties of radiation mapping. To tackle this challenge, we further study the radiation distribution map, which includes multiple radiation sources (Fig. 4). We can see good agreement can be achieved for two radiation sources. However, we would like to point out that more detector pixels such as 10 × 10 (Fig. 4a) or 5 × 5 (Fig. 4b) grids are used since the 2 × 2 square-grid detector does not show adequate performance unless the restriction of fixed distances between the detector and the radiation sources, as shown in Fig. S6. Figures S13 and S14 in Supplementary Note 5 present the intermediate process of radiation mapping with the square detectors of 10 × 10 and 5 × 5 configurations, respectively.

**Experimental validation of radiation mapping with a 4-pixel detector**

In the preceding sections, we demonstrated the efficacy of our machine-learning approach in accurately locating radiation sources in 2D space using MC simulation data. To validate the practical utility of our method in the field of radiation measurement, it is essential to assess its performance in real-world experimental scenarios.

We conducted a comprehensive experiment to map the location of a radiation source within a real-world environment. As Fig. 5a shows the experimental schematics, the measurement involved positioning a Cs-137 radiation source at coordinates (5.0, 0.0, 0.0), while the experimental team kept the source position secret until radiation mapping algorithms predicted it. We deployed a radiation detector configured in a 2 × 2 square layout and moved the detector around the area near the radiation source. The detector outputs the radiation absorption by each crystal (pixel) at regular intervals of 0.5 s. We employed a signal smoothing technique to reduce measurement fluctuations. "Methods" and Supplementary Note 7 explain further details regarding the experimental setup and data analysis, including a conventional non-neural-network approach for radiation source mapping.

As an existing analysis method, we demonstrated the non-ML gridded point source likelihood (GPSL) reconstruction method[7] for comparison. Figure 5b shows a top-down view of the measurement and GPSL reconstruction for the measurement. The gray points are the LiDAR point cloud of the scanned area, and the red, green, and blue lines show the detector's $x$, $y$, and $z$ axes at each 0.5 s timestamp. The path between the axes is colorized by gross counts (qualitatively) from low (cyan) to high (magenta). The color bar denotes the likelihood contours of $z$-scores up to $5\sigma$ that the given 10 cm pixel contains a point source. The most likely point source position is highlighted by the black dashed crosshair, while the red dashed crosshair shows the actual source position. With the limited approach to the source afforded by the circular detector trajectory, there is a slight 0.75 m error in the reconstructed position, but the activity estimate closely matches the actual value of 170.8 $\mu$Ci.

In Fig. 5c–f, we present the results of our MAP analysis applied to radiation mapping with the 4-pixel detector. The maps depict probabilities of holding a radiation source. The area denoted by intense red shading represents the higher probability, which precisely converges around the ground-truth position of the radiation source marked by "×." This convergence underscores that our neural network, equipped for directional prediction and MAP analysis, effectively approximated the actual location of the radiation source with equivalent quality to the GPSL approach. We provide additional images offering both top-down ($z$-axis direction) and aerial perspectives in Figs. S21 and S22 of Supplementary Note 7 for a comprehensive view of the mapping process at various time intervals. In Supplementary Movie 11, we present the measured signal at each timestamp and the process by which the moving detector maps the radiation in the experimental scenario.

## Discussion

The conventional detector configuration has a grid structure vertically facing the source of detection, where each detector pixel receives the radiation signal with a slightly different solid angle. In this work, we propose an alternative detector configuration with a few features. First, the detector grid is placed horizontally within the plane instead of vertically facing the source. Second, additional thin padding layers are padded between detector pixels, i.e., the contrast between pixels is not only created by incident angles but also enhanced by padding layers that are good absorption layers of radiation. Third, machine learning algorithms are implemented to analyze the detector reading, demonstrating great promise to reduce the need for detector pixel numbers and thereby reduce the cost of fabrication and deployment. Fourth, non-conventional Tetris-shaped detector geometry is proposed beyond the square grid, which can lead to more efficient use of pixels with improved resolution, particularly for the S-shaped Tetris detector. Finally, we demonstrate the possibility of locating the positions of radiation sources in a moving detector scheme through MAP. Experimental validation could further prove the capability of our machine learning approach for locating radiation sources in the real-world scenario.

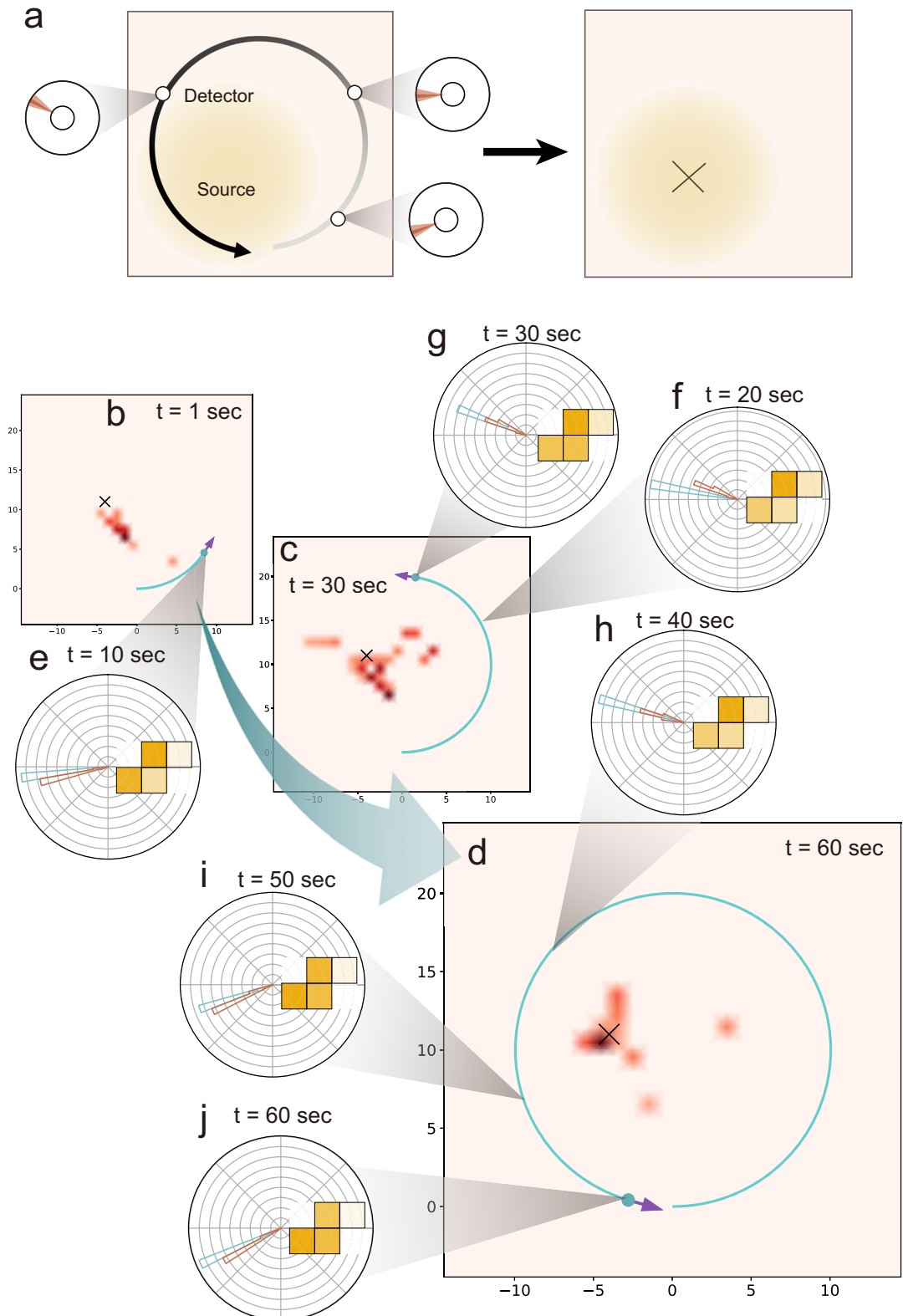

**Fig. 3 | Radiation mapping and position determination with an S-shape Tetris-inspired detector. a** By acquiring detector readings at each spatial position during the circular motion, the position of the radiation can be gradually optimized through MAP. **b**–**d** The process to map the radiation source at a few representative times at $t$ = 10, 30, and 60 s, respectively. The "×" symbol on the maps shows the ground-truth location of the radiation source. The areas colored with intense red indicate a high probability of where the radiation source is located. The purple arrows indicate the front side of the radiation detector. **e**–**j** The detector's input signals and the predicted directions of the radiation sources at time $t$ = 10, 20, 30, 40, 50, and 60 s. Both the input signal and the curves of the polar coordinates are visualized in the detector frame. The top side of the detector represents the front side. Check the radiation mapping process in Supplementary Movie 1.

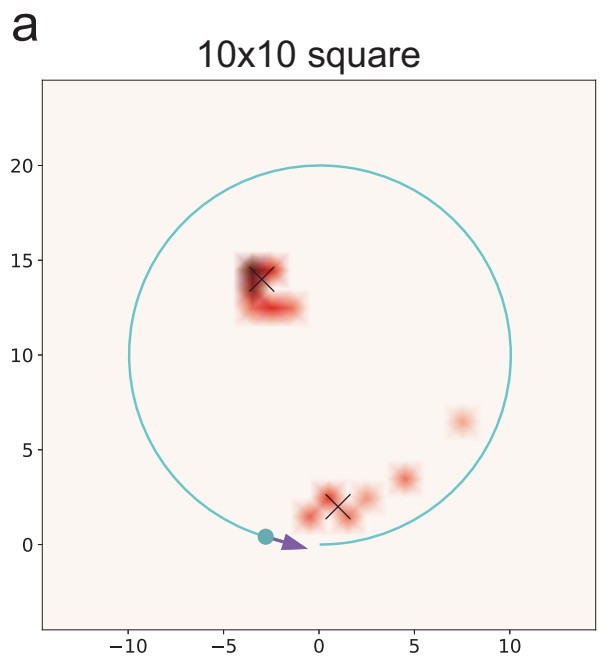

**Fig. 4 | Radiation mapping of two radiation sources with a moving detector.** Two radiation sources are placed in the space (shown as the two black crosses of "×"). The detector is moving in a circular motion around the sources (blue circles).

We use the detector of $10 \times 10$ grid (in **a**) and $5 \times 5$ grid (in **b**). Check the radiation mapping process in Supplementary Movies 5 and 6.

Despite these initial successes, we believe the configuration proposed in this work is still in its infancy. Several refined works are foreseeable. Particularly, although the 2D configuration can represent several realistic scenarios, such as radiation sources are far away from the detector but still close to ground level, it would still be an interesting problem to study 3D configuration, possibly with 3D detectors like Rubic-shaped detector cubes. Moreover, several improvements, such as moving radiation sources and energy spectra of radiation, may be feasible with a more advanced approach like reinforcement learning. Our work represents one step that leverages the detector pixels and shapes with machine learning toward radiation detection with reduced complexity and cost.

## Methods
### Monte Carlo simulation of static detector and data representation
The training data, in other words, the intensity measured by each detectors is simulated by MC Simulation based on the principle of radiation-matter interaction. We used an existing MC simulation package called OpenMC[28]. OpenMC incorporates effects like Compton scattering and pair production, enabling simulations to accurately model radiation interactions in realistic scenarios. Some representative results of the detector arrays is shown in Fig. 2. For the sake of simplicity, we temporarily assume the radiation source and the detector arrays are in the same plane. In the MC simulation, first we define the geometry of the detector. Schematic figures of detectors arrays are shown in Fig. 1. The adjacent detectors (yellow) are separated by attenuation materials (black), which forms the detectors' configuration like lattices. We set the distance $d$ (cm) between the center of the detector and the radiation source. The direction of the radiation source is defined as an angle $\theta$, which is defined in clockwise direction from the front side of the detector. When we generate training data, we selected $d$ and $\theta$ at random ($d \in [20, 500]$, $\theta \in [0, 2\pi)$) so that the neural network could get feature from radiation sources of various distances and directions. The distribution of the radiation source positions is shown in Fig. S1. After MC simulation was

completed, the detector's readouts are represented as the matrix of ($h \times w$), where $h$ and $w$ are detectors' dimensions of heights and width respectively. For the square detector comprised of four detector panels, the data of $2 \times 2$ matrix was normalized so that the mean and the standard deviation are 0 and 1, respectively. For the detectors of Tetromino-shapes, the detectors' readouts are represented as $2 \times 3$ matrices. Since two sites of the matrices' 6 elements are vacant, we filled them with zero and did normalization in the same way as the square detectors. We followed the same MC simulation method as above to generate the 64 filter layers, which we explain in more detail in this section. All other parameters used in MC simulations are shown in Table S1.

The dataset $D$ is in the form of $\{\mathbf{x}^{(i)}, \mathbf{y}^{(i)}\}_{i \in [1, N_1]}$, where $N_1$ is the size of the dataset. $\mathbf{x}^{(i)} \in \mathbb{R}^{h \times w}$ is the normalized readouts of the detector arrays h, w denotes the number of rows and columns of the detector arrays, respectively. For example, $h = w = 2$ for $2 \times 2$ square detector, $h = 2$, $w = 3$ for Tetromino-shape detector. $\mathbf{y}^{(i)} \in \mathbb{R}^{N_a}$ is the angular distribution of the incident radiation, $N_a$ is the number of sectors that are used to separate $[0, 2\pi)$. Each element in $\mathbf{y}^{(i)}$ represents the ratio of incident radiation intensity received from the direction of this sector to the total incident radiation intensity. For the point sources, the angular distribution of the radiation source is represented by the following method. For an angular distribution $\mathbf{y}$ contributed by multiple point sources, let $\mathbf{y}_j$ represent the angular distribution contributed by the $j$th point source. The $k$th element of $\mathbf{y}_j$ is defined by:

$$y_{jk} = \begin{cases} 0, & \text{if } |\theta_j - \frac{2\pi}{N}(k-1)| \geq \frac{2\pi}{N} \\ \frac{|\theta_j - \frac{2\pi}{N}(k-1)|}{\frac{2\pi}{N}}, & \text{else} \end{cases} \quad (1)$$

We also have:

$$\mathbf{y} = \sum_j \frac{I_j}{I_0} \mathbf{y}_j \quad (2)$$

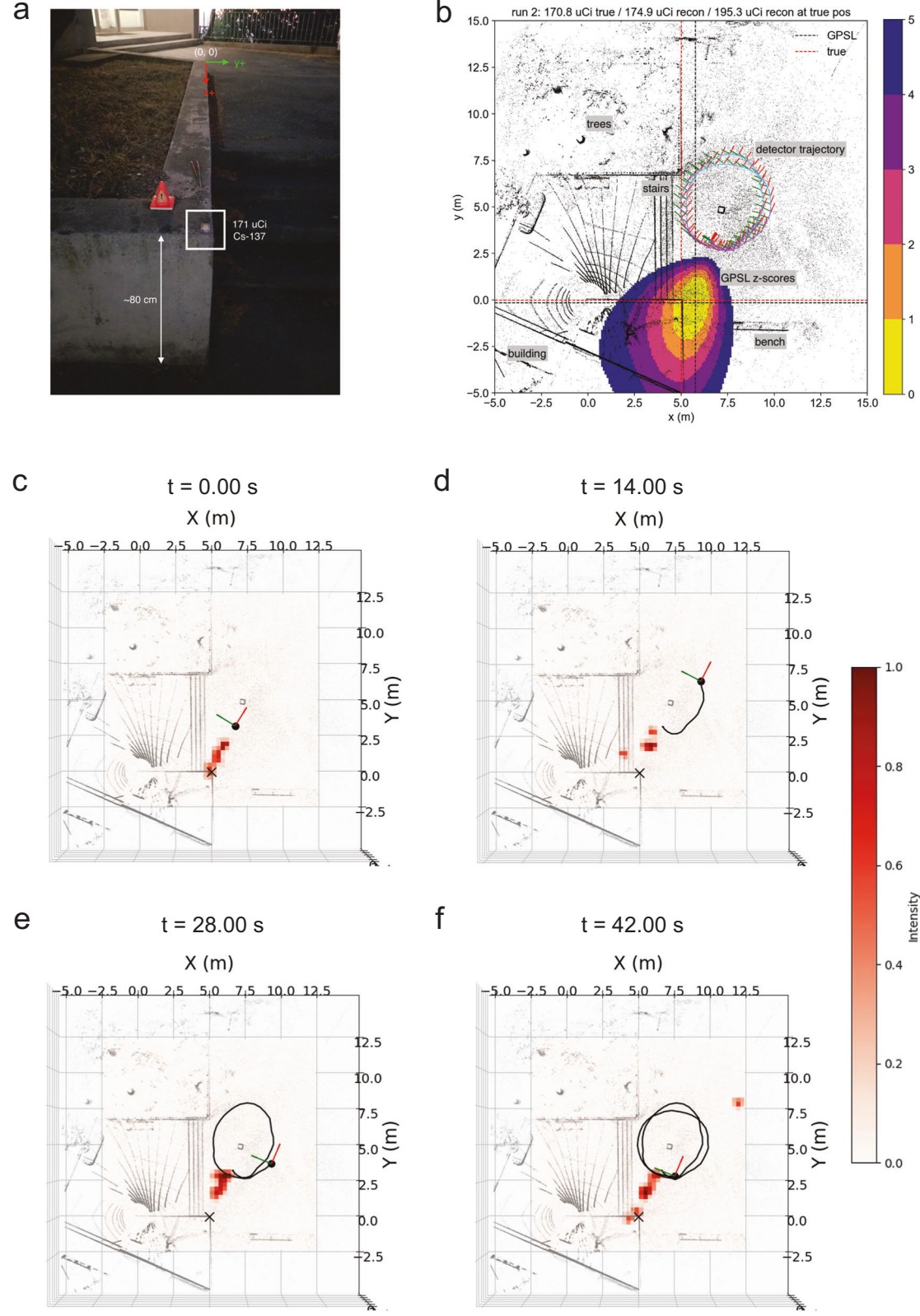

where $\theta_j$ denotes the incident angle of the $j$th radiation source, $I_j$ denotes the total incident intensity revived from the $j$th radiation source, $I_0 = \Sigma_j I_j$ s the total incident intensity revived from all radiation sources. By using this representation, we are able to accurately represent the incident direction of a point source with an arbitrary angle, with a discretized angle interval. In the experiments, $[0, 2\pi)$ is separated in to $N_a = 64$ sectors. Figure 2 illustrates this representation by a pie chart.

## Deep neural network architecture
In order to extract the global patterns of the input data, a set of global filters is designed. We obtain several filters based on very high-quality

**Fig. 5 | Experimental setup and radiation mapping with experimental measurement data (top view). a** Annotated photograph of the measurement setup. The red and green arrows show the approximate $x$- and $y$-axes of the detector coordinate, and the 171 $\mu$Ci Cs-137 source is shown on the corner of the concrete ledge about 80 cm above the sidewalk level. Note that the radiation source was deliberately positioned at coordinates (5.0, 0.0, 0.0), and the data analysis remained intentionally blinded to the true source location until they made a prediction. **b** Top-down view of the GPSL reconstruction $z$-scores of the measurement. The thin bands of black LiDAR points appearing around (0.0, 0.0, 0.0) are artifacts from the system's initial static dwell. **c–f** The progression of radiation source mapping at representative time intervals of $t = 0, 14, 28, 42$ s, respectively. The gray point clouds in the diagrams represent the surrounding environment of our experiment. The symbol "×" designates the ground-truth location of the radiation source. The black dot on the maps indicates the position of the radiation detector. We visualize the left ($y$) and front ($x$) axes of the detector with green and red arrows, respectively. The black solid line indicates the trajectory of the detector. Check the radiation mapping process in Supplementary Movie 11.

simulations in some representative cases, including far-field incident (the radiation source is located at a far distance compared to the size of the detector arrays) and near-filed (the radiation source is located at a close distance) incident at different directions. As an example, near-field filters of the S-shape detector are shown in Fig. S2. Each filter has the size of (h × w), which is set same as that of the training data. The weight of each unit is given by the readout of each single pad detector in the simulation. The output of this layer is given by:

$$Z_{mk} = \sum_i \sum_j x_{ij} w_m F_{mkij} + b_{mk} \tag{3}$$

where $Z_{mk}$ is the $k$th element of the output array at channel $m \in \{1, 2\}$. Channel 1 and 2 correspond to the far-field and near-field filters respectively. $x_{ij}$ denotes the input array at pixel $i, j$, $F_{mkij}$ denotes the $(i, j)$ element of the global filter obtained in the case that incident radiation from the $k$th sector, in far-field ($m = 1$) or near-field ($m = 2$) scenario. $w_m$ denotes a channel-wise normalization weight, and $b_{mk}$ denotes the bias for this global filter. During training, the weights of the global filter are initialized with the far-field and near-field filters that we obtained from the high-quality simulations. The weights are slightly fine-tuned with a learning rate lower than the learning rate of the other layer of the network, while the bias for each filter is trained with the same learning rate as the other layers. The filter layer is followed by an Exponential Linear Unit (ELU) activation function[29]. The output of the global filtering layer embeds the directional information corresponding to the direction of the filter channel. It is then fed into the U-shape network to extract the directional information.

In the neural network, the input data is normalized (h × w) detectors readout. However, it is essentially different from images captured by cameras. For the processing of images which is measured from visible light, convolution neural networks (CNNs) are widely used[30]. A convolution layer is used for extracting features that are presented as localized patterns. However, due to the penetration properties of several kinds of radiation, such features are presented as global patterns, which is different from the imaging of visible light. Therefore, an updated architecture is designed for this purpose; the input data is followed by a global filtering layer with the shape (2, 64, $h$, $w$) in order to extract the global pattern. The output of this layer conveys the directional information with a size of (2, 64), which corresponds to the final output, i.e., the estimated angular distribution, with a size of (1, 64). In order to perform a pixel-to-pixel prediction of the angular distribution, a U-shape fully convolutional architecture which is similar to U-Net[31] is then utilized as Fig. 1b–d shows. Noticed that the U-Net architecture was originally developed for image segmentation[31]. However, here the output is a 1D array, and the input is viewed as a 1D array with two channels. Thus, we accordingly set the dimension of the U-Net, as is shown in Fig. S3. Finally, the output of the final layer feeds into a softmax layer for normalization.

In order to represent the distributional similarity between the predicted and target distribution, Wasserstein distance is proposed to be used as the loss function. It is a distance function on a given metric space between two probability distributions. As this metric is an

analogy of the minimum cost required to move a pile of earth into the other, it is also known as the earth mover's distance[32–34]. The Wasserstein loss function is given by:

$$W = \min_{\pi(i,j)} \sum_i^n \sum_j^n \pi(i,j) C(i,j) \tag{4}$$

$$\begin{aligned}
\text{subject to} \quad & \pi(i,j) \geq 0, && \forall i,j \in [1, N] \\
& \sum_j \pi(i,j) \leq y_i, && \forall i \in [1, N] \\
& \sum_i \pi(i,j) \leq \hat{y}_j, && \forall j \in [1, N] \\
& \sum_i \sum_j \pi(i,j) = 1
\end{aligned} \tag{5}$$

where $\pi(i,j)$ is the transport policy which represents the mass to be transferred from state $i$ to state $j$, $y_i$ is the ground truth of the normalized angular distribution, $\hat{y}_i$ is the estimated angular distribution. $C(i,j)$ is the cost matrix representing the cost of moving unit mass from state $i$ to $j$. Our work is in a cyclic case and uses the following form:

$$C(i,j) = \min(|i - j|, |j + n - i|, |i + n - j|)^l \tag{6}$$

An algorithm is developed to calculate the cyclic Wasserstein distance, as is shown in Algorithm 1. Particularly, the cyclic case is unrolled into an ordered case. The ring is split into a line at $n$ different units and obtain $n$ different distributions. The cost matrix in the cyclic and ordered cases. The Wasserstein distance could be computed in closed form[33,35]:

$$W(p,t) = \left(\frac{1}{n}\right)^{\frac{1}{l}} \| CDF(p) - CDF(t) \|_l \tag{7}$$

Where $CDF(\cdot)$ calculates the cumulative distribution of its input. Following this formula, a decycling algorithm is developed to calculate the Wasserstein distance with a cyclic cost. The algorithm is shown below:

**Algorithm 1. Cyclic Wasserstein Distance**

**Input:** Array $p$ and $t$ of size $N$,
**Output:** Wasserstein Distance: $Dist$
$\mathbf{d} \leftarrow$ new array of $N$
**for** $i \leftarrow 1$ **to** $n$ **do**
 $p_{\text{new}} \leftarrow$ concat($p[i+1 \text{ to } N], p[1 \text{ to } i]$)
 $t_{\text{new}} \leftarrow$ concat($t[i+1 \text{ to } N], t[1 \text{ to } i]$)
 $\mathbf{d}[i] \leftarrow \left(\frac{1}{N}\right)^{\frac{1}{l}} ||CDF(p) - CDF(t)||_l$
**end**
$Dist \leftarrow \min(\mathbf{d})$

It could be numerically verified that this algorithm enables exact calculation of 1st ($l = 1$) Wasserstein distance for the cyclic Wasserstein distance, given arbitrary distribution. This algorithm is differentiable and enables us to optimize the objective through back-propagation. As

for evaluation in the experiments, we use 1st Wasserstein distance which can directly represent the angle difference of the estimated and real directions. In the training process, we use the above 2nd ($l = 2$) Wasserstein distance as a loss function since it usually converges faster with gradient descent-based optimization methods, compared to the 1st Wasserstein distance[33,36].

In the proposed model, the network is trained using Adam[37] with a learning rate of 0.001 for all parameters in the neural network except for the weights of the global filtering layer, whose learning rate is set to $3 \times 10^{-5}$. The training batch for each step is randomly selected from the training set which is based on several pristine simulation result set $D^{\text{train}}$ on one radiation source. All models randomly split the data into 90% training (2700 data) and 10% testing (300 data) sets. Furthermore, we trained our models with a 5-fold cross-validation scheme. We summarize the parameters for training the neural network for predicting the directions of one or two radiation sources in Table S2.

### Radiation source mapping with maximum a posteriori (MAP) estimation

We set up a radiation mapping problem by considering the case that there is one point source in the environment. This task could be extended to a Simultaneous Localization and Mapping (SLAM) problem[15]. The directional radiation detector could be viewed as a kind of sensor that could only obtain directional information, and a radiation source could be viewed as a particular landmark that could only be measured by this kind of detector. We show that by treating the directional radiation detector as a sensor with only directional resolution, it could be easily integrated into the MAP optimization framework and it enables source localization. Details of how to integrate the directional radiation detector into the MAP framework are presented in Fig. S8.

Here, a method based on Maximum a Posteriori (MAP) Estimation is proposed in order to generate the radiation distribution map. In addition, here, we assume that the carrier for the detector has already localized itself and is only required to build the radiation map. The map is discretized into a mesh with $N_m$ square pixels. Let $c \in \mathbb{R}^{N_m}$ denote the radiation concentration at each pixel. It could be assumed that the radiation is uniformly generated from the pixel. The measurements result $\mathbf{z}_t = I_0 \mathbf{y}_t$ denotes the incident radiation intensity coming from different directions at time $t$. It could be assumed that the measurement probability $p(z_t|c)$ is linear in its arguments, with added Gaussian noise:

$$\mathbf{z}_t = \mathbf{M}_t \mathbf{c} + \boldsymbol{\delta} \tag{8}$$

where $\delta \sim N(0, \Sigma_\delta)$ describes the measurement noise, $\mathbf{M}_t \in \mathbb{R}^{N \times N_m}$ denotes the observation matrix at time $t$. Note that in our study, we treat gamma measurements as continuous real numbers with Gaussian noise, which is appropriate for scenarios where the sample size is sufficiently large, and the mean count rate is not significantly low. The central limit theorem ensures that for a significant sample size, the Poisson distribution, which describes the discrete nature of event counts, tends to approximate a Gaussian distribution. As such, our choice of Gaussian distribution provides a reasonable approximation for the behavior of directional radiation detectors under the conditions of our experiments, where the counts are reasonably large, allowing us to accurately model the measurement uncertainties. Considering the contribution of one pixel to one direction sector of the detector, only the overlapped area can contribute to the sector, as is shown in Fig. S8, and the intensity is proportional to the overlapped area. Besides, the intensity is inversely proportional to the square of the distance between the detector and the source. Therefore, the element of $M_t$, or in other words the intensity contribution of the $i$th pixel to the $j$th directional sector of the detector at time $t$ can be written as:

$$M_{tij} = \frac{A_{tij}}{r_{ti}^2} \tag{9}$$

where $A_{tij}$ denotes the area of the overlapped region of the $i$th pixel and the $j$th sector at time $t$ (blue area in Fig. S8), $r_{ti}$ denotes the distance between the detector and the center of the pixel. According to Bayes' rule, we have:

$$
\begin{aligned}
p(\mathbf{c}|\mathbf{z}_1, \mathbf{z}_2, \ldots, \mathbf{z}_t) &= \frac{p(\mathbf{z}_1, \mathbf{z}_2, \ldots, \mathbf{z}_t|\mathbf{c})p(\mathbf{c})}{p(\mathbf{z}_1, \mathbf{z}_2, \ldots, \mathbf{z}_t)} \\
&\propto p(\mathbf{z}_1, \mathbf{z}_2, \ldots, \mathbf{z}_t|\mathbf{c})p(c) \\
&= p(\mathbf{c}) \prod_{i=1}^{t} p(\mathbf{z}_i|\mathbf{c})
\end{aligned} \tag{10}
$$

Here, we assume that measurements at different times are conditionally independent given $\mathbf{c}$. As we are trying to find $\mathbf{c}$ that maximizes $p(\mathbf{c}|\mathbf{z}_1, \mathbf{z}_2, \ldots, \mathbf{z}_t)$, the $p(\mathbf{z}_1, \mathbf{z}_2, \ldots, \mathbf{z}_t)$ term could be ignored as it is independent of $\mathbf{c}$. It could be assumed that the prior term $p(\mathbf{c}) = N(0, \varepsilon I)$ is a Gaussian distribution. Then following the Maximum a Posteriori (MAP) estimation, we have:

$$
\begin{aligned}
\underset{\mathbf{c}}{\text{argmax}}\ p(\mathbf{c}|\mathbf{z}_1, \mathbf{z}_2, \ldots, \mathbf{z}_t) &= \underset{\mathbf{c}}{\text{argmax}} \sum_t \ln p(\mathbf{z}_t|\mathbf{c}) + \ln p(\mathbf{c}) \\
&= \underset{\mathbf{c}}{\text{argmin}} \sum_t \| \mathbf{M}_t \mathbf{c} - \mathbf{z}_t \|_{\Sigma_\delta}^2 + \frac{1}{\varepsilon^2} \| \mathbf{c} \|_2^2
\end{aligned} \tag{11}
$$

Therefore, the radiation concentration distribution could be obtained by solving the optimization problem:

$$\min_{\mathbf{c}} \sum_t \| \mathbf{M}_t \mathbf{c} - \mathbf{z}_t \|_{\Sigma_\delta}^2 + \frac{1}{\varepsilon^2} \| \mathbf{c} \|_2^2 \quad \text{subject to}\ \ c_i \geq 0,\ \forall i \in [0, N_m] \tag{12}$$

where $\frac{1}{\varepsilon^2} \| \mathbf{c} \|_2^2$ term could be viewed as a regularization term. If we do not have much information regarding the prior distribution, $\varepsilon$ will be a large number. This term will penalize large concentration if the measuring data is inadequate to determine the concentration (i.e., the area is not fully explored and caused very small $M_{tij}$). In practice, $\varepsilon$ could be tuned by utilizing differentiable convex optimization layers[38], in which the optimization problem could be viewed as a layer within the neural network, and error back-propagation is enabled through implicit differentiation, given predicted and ground truth data. In our demonstration case, we simply set $\frac{1}{\varepsilon^2} = 0.1$ such that the regularization term is relatively small.

### Experimental setup of real-world radiation mapping and post-process for MAP analysis

The system comprises a $2 \times 2$ array of $1 \times 1 \times 2''$ CLLBC ($Cs_2LiLa(Br,Cl)_6$:Ce) gamma/neutron detectors separated by a poly-ethylene cross. The detector is equipped with a Localization and Mapping Platform (LAMP) sensor suite used to map the 3D environment and determine the detector pose (position and orientation) within that environment, and the demonstration measurements were made with Lawrence Berkeley National Laboratory's Neutron Gamma LAMP (NG-LAMP) radiation mapping system[39]. A 171 $\mu$Ci Cs-137 check source was placed on a concrete ledge in an outdoor environment, and the detector system was used to make free-moving measurements of the source. The detector was hand-carried in a circular pattern of about 2 m radius around a point 5 m away from the source location for up to 45 s, completing almost 2.5 loops. Throughout the measurement, the detector orientation was kept almost fixed with respect to the environment. The CLLBC crystals were kept at a height close to that of

the radiation source during the entire measurement duration. The listmode radiation data in an energy region of interest (ROI) of [550, 800] keV and the detector pose determined by NG-LAMP's contextual sensor suite were then interpolated to a 0.5-s time binning. We note that these real-world radiation measurements placed the source outside the circular detector trajectory to model a realistic source-search scenario closely.

To enhance the quality of measurement data for analysis, we applied a series of post-processing steps to the measured data. First, the location of the 661.7 keV Cs-137 photopeak in each crystal had drifted to 690–700 keV, necessitating a manual, multiplicative gain correction for each crystal. To avoid any possible energy aliasing from this gain correction, the corrected energy values were blurred by a Gaussian kernel of standard deviation 1 keV, much smaller than the width of the photopeak (10 keV). Furthermore, other post-processing steps were applied to the contextual data to streamline the analysis. The light detection and ranging (LiDAR) point clouds and global coordinate frames were more precisely aligned to a single coordinate frame using the Iterative Closest Point (ICP) algorithm in Open3D[40,41]. Moreover, the initial and final parts of the measurements where the system was walked to/from its intended measurement position(s) and used to perform a dedicated LiDAR scan of the area were cut from the radiation mapping analysis. The trajectory and radiation data were cut here, but the contextual LiDAR point clouds were not.

As an existing analysis method for comparative analysis, We demonstrated the non-machine-learning gridded point source likelihood (GPSL) reconstruction method[7]. Using quantitative response functions, GPSL computes the best-fit source activity for every potential source point in the imaging space and selects the source point with the maximum likelihood[16]. As in the neural network analysis, reconstructions were computed for the 2D plane level with the actual source height, using an energy region of interest (ROI) of 661.7 ± 80 keV.

We conducted radiation mapping to validate that our neural networks and MAP are applicable to the radiation measurement in a real-world scenario. As detailed below, we first pre-processed experimental data as the inputs of our analysis. At each position of the 91 timestamps through 45-s measurements, the $2 \times 2$ square detector acquired measurement data as a matrix of dimensions (4, 84). These measurements corresponded to the count of photons absorbed by the four pixels on the crystal panels. The photon counts were recorded for 84 energy bins ranging from 550 to 800 keV. To process this raw measurement dataset into the input dataset of our neural network model, we summed the photon counts absorbed by each pixel across this entire energy region of interest.

The initial detector signals exhibited significant statistical fluctuations, making it challenging to predict the radiation source direction accurately. When the detector remained stationary at the starting pose, the radiation source direction predicted by our algorithm changed abruptly in the early timestamps. To mitigate this issue, we implemented a signal-smoothing technique using a moving average filter written in Eq. (13). This process smoothed the detector's signals by averaging the signals of the neighboring $2M+1$ timestamps. We used $M$, equal to 3, resulting in a window size of 7 timestamps. This smoothing technique reduced the total number of timestamps from 91 to 85. The smoothed signal data served as the input for our machine-learning model:

$$\mathbf{x}'_t = \frac{1}{2M+1} \sum_{i=-M}^{M} \mathbf{x}_{t+M+i} \tag{13}$$

We employed a U-Net architecture trained using MC simulation data to predict the direction of the radiation source given the signals from the detector of $2 \times 2$ configuration. The trained model was then applied to predict the radiation direction based on the smoothed measurement data. With directional information predicted by our model at each timestamp, we applied the MAP method to reconstruct the radiation map. We restricted the MAP analysis to a $15 \times 15$ square area within the entire space to reduce computational complexity.

## Data availability

The Supplementary Movies showing the radiation mapping processes are available. The training data generated with the OpenMC[28] package have been deposited in the GitHub repository (https://github.com/RyotaroOKabe/radiation_mapping/tree/main/save/openmc_data/saved_files).

## Code availability

The source code is available at https://github.com/RyotaroOKabe/radiation_mapping.git[42]. The GitHub repository and Supplementary Note 8 present the instructions for reproducing the results of our simulations and machine learning.

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

## Acknowledgements

R.O. and M.L. thank the helpful discussions from F. Frankel. R.O., T.L., G.K., and M.L. acknowledge the support from U.S. Department of Energy (DOE), Advanced Research Projects Agency-Energy (ARPA-E) DE-AR0001298. R.O. and M.L. are partly supported by DOE Basic Energy Sciences (BES), Award No. DE-SC0021940. R.O. acknowledges support from Heiwa Nakajima Foundation. M.L. acknowledges the Norman C Rasmussen Career Development Chair, the MIT Class of 1947 Career Development Chair, and the support from Dr. R. Wachnik. This manuscript has been authored by an author at Lawrence Berkeley National Laboratory under Contract No. DE-AC02-05CH11231 with the U.S. Department of Energy. The U.S. Government retains, and the publisher, by accepting the article for publication, acknowledges, that the U.S. Government retains a non-exclusive, paid-up, irrevocable, world-wide license to publish or reproduce the published form of this manuscript, or allow others to do so, for U.S. Government purposes.

## Author contributions

R.O. took the lead in the project. R.O. and S.X. developed the machine learning model. R.O. performed the Monte Carlo simulations with support from T.L., J.Y. and B.F. R.O. wrote the manuscript with support from S.X. and J.V. J.V., R.P., V.N., B.Q. and J.C. carried out the experiment. R.O., S.X. and J.V. carried out data analysis. M.L., S.J., G.K. and L.H. designed the project, provided supervision, and revised the manuscript.

## Competing interests

The authors declare no competing interests.
