## [Peer Review File · Nature Communications]

Tetris-inspired detector with neural network for radiation mappingREVIEWER COMMENTS

Reviewer #1 (Remarks to the Author):

The authors had a simulation study of Tetris-shaped detector with CZT using a neural network to improve the direction of gamma-ray (x-ray). It is crucial to prove authors simulation study with experimental data since the authors Monte Carlo (MC) simulation does not include detector effects such as noise and resolution effects, typically MC all the time to give better results than experimental data. Also it is necessary to compare the author's results with other direction and position finding methods and prove author's results are better than others. Since it lacks of the above study, I don't recommend this manuscript for publication in nature communication.

The following a major comment

1. It is necessary to do the experiment with CZT and radio-active gamma-source to prove that the authors MC study shows consistent results with the data
2. To find the direction of gamma-ray (X-ray), Coded Mask and Compton Camera methods are popular. Thus authors need to compare with author's results with the above method to prove that the authors method is better than the other methods.
3. It is better to show angular and position resolution quantitatively with different configurations and minimum data requirements for imaging. Also, the authors need to show gamma-ray energy dependency of the position and angular resolution.
4. Abstract : to locate the directions and position of the radiation sources. → locate the directions and position of the gamma-ray (X-ray) sources.

Reviewer #2 (Remarks to the Author):

The authors report a simulation of idealized radiation detectors in two dimensions and propose a means by which a detailed map of the radiation environment may be obtained using only four pixels arranged in a few simple configurations. The work is important as while it is a straightforward exercise to sense the presence of a radioactive source, determining its direction or location must be determined with grid searches or vast arrays of detectors. Either approach has its drawbacks as they are either labor intensive or expensive, requiring large arrays of sensors or elaborate masking schemes and post processing.

The authors' monograph demonstrates that by translating the sensor array around the area under study and applying neural network filters, it is possible to map out the radiation sources in that area with some fidelity. Rotating the sensor arrays improves the fidelity substantially.

The work is purely simulation and begs validation experimentally. An example which could be performed in the real world is to map out a space using cylindrical detectors arranged in the authors' proposed arrangements. At least within the plane of the experiment results could be drawn which can be correlated with reality, as opposed to a simulation.

The authors' language is good but not perfect, with some errors in agreement of number and object. And several acronyms or proper names are uncapitalized when they should be. Even so, the small imperfections in grammar are not a distraction from the intent of the text. The writing is fairly clear, though the benefit of the animations included with the text make the meaning much more apparent.

I believe this work should be accepted. As the only problems I find are with language I leave it for you the editor to determine whether the authors or your proofreaders are the ones to tighten up the text.

Reviewer #3 (Remarks to the Author):

Using self-shielding/shadowing effect for detecting the direction of gamma or neutron sources is not completely new. However, the way of doing the mapping with neural network (for which I am not an expert) seems original and valuable. I have the following comments:

1) The title is about radiation mapping, it is better to cite a few neutron localisation studies. Here is one: https://indico.cern.ch/event/731980/contributions/3285768/attachments/1829613/2995886/VACHARE_T_NuSec-Poster-2017.pdf

2) In the last paragraph of Introduction section, you should state that the work is for gamma radiation only. The penetrating/scattering trajectory and detection mechanism of neutrons are totally different from the work presented here.

3) In line 67 it says "the 5 meters of source-detector distance". But in your Supplementary Information, Figure S1, 5 meters is the upper limit of the source-detector distance. The lower limit of the distance is 0.2 meter (Table S1). You used phrase "near-field" quite a few times, that means the source can be very

close to the detector. I think, including “near-field” scenario may be not beneficial. I will talk about this later in my comments.

4) In Figure 1d, it is not clear why Simulation and Prediction have more than one sections of different shade. I guess this is the mapping from source pixels to detector sectors. Will these shaded sections come to 1.0 or 100% when added together quantitatively? You may want to explain.

5) About Table 1, one would think for each type of Tetrominos the sensitivity around 360 degree angle is not even. At some angles, the angle change will cause more significant change in the shadowing effect, yet at some other angles, less significant. I suggest to change Table 1 into 3 rows, Error at most favourite angle, Error at least favourite angle, Error average (maybe this is the one in your table now).

6) In line 109, time steps are introduced. In my opinion, time is meaningless without talking about source intensity, in Bq, or in photon/s. In Table S1, the number of photon particles are given (50000, which seems too small based on my limited MC simulation experience) but it does not help in determining the source intensity. You need a time interval, which is independent of MC simulation.

7) In line 122, you stated that 2x2 square-grid cannot detect 2-sources in the view. Do you mean only the square-grid, or also include the S- L- and T-shaped grid? In my opinion, this will fail if you want to distinguish near- and far-field. For one source, you need to figure out 1) the intensity, 2) the angle, and 3) the distance, your measurement gives you 4 detector readings, so you are able to solve 3 variables (using neural network is the same thing). For two sources, you need to solve 6 variables with 4 detector readings, which is impossible, here neural network will not help. However, if you only consider far-field, say training your system at a fixed source-detector distance, you may get the location of two sources (without trying to solve the distances) because the angle information will pinpoint the locations.

8) About Equation (6), here lies the biggest concern of mine. You seem to treat gamma measurement as continuous real number, can be infinitely small (or large), with Gaussian noise. In reality, the measurement follows Poisson distribution. You can never get 1.83 counts, but you may get 0, 1, 2, 3, and so on. Poisson distribute does gradually become close to Gaussian distribution, but it requires big counting numbers (higher source intensity, longer measurement time, less shielding effect from padding). I am not sure you have taken this into consideration.

9) Are you able to deal with background counts? Subtracting background (which also follows Poisson distribution) will introduce more uncertainty.

10) A comment about the gamma measurements, in line 69 and line 72 you stated assuming 0.5 MeV, and that “only counting matters”. These are OK for the initial study, but there will be problems in reality. Because gamma ray can go through Compton scattering and pair-production, the nearby structure (including the items beside and behind the detector) may affect the counts registered as 0.5 MeV gamma. This may cause the trained neural network to become incorrect as the detector start to rotate and travel as shown in Figure 3a.

11) Minor issues

a. In line 44, “single sources” should be “single source”.

b. In line 68/69, “but not absorb the photons in the gamma range” is not correct, it should be about the degree of absorption, not a complete “no”.

c. In Figure 1a I found that the arrows are misleading. I prefer to have them removed.

d. In Figure 1d and many other locations, the shaded sectors can be replaced with step-curves in polar coordinates. This will be helpful for color-blinder readers like me.

e. In Supplementary Information, line 17/18, Pd should be Pb.

Responses to Reviewers' Comments for Manuscript

Tetris-inspired detector with neural network for radiation mapping

Addressed Comments for Publication to

by

Ryotaro Okabe, Shangjie Xue, Jayson R Vavrek, Jiankai Yu, Ryan Pavlovsky, Victor Negut, Brian J Quiter, Joshua W Cates, Tongtong Liu, Benoit Forget, Stefanie Jegelka, Gordon Kohse, Lin-wen Hu, Mingda Li

Authors' Response to Reviewer 1

General Comments. The authors had a simulation study of Tetris-shaped detector with CZT using a neural network to improve the direction of gamma-ray (x-ray). It is crucial to prove authors simulation study with experimental data since the authors Monte Carlo (MC) simulation does not include detector effects such as noise and resolution effects, typically MC all the time to give better results than experimental data. Also it is necessary to compare the author's results with other direction and position finding methods and prove author's results are better than others. Since it lacks of the above study, I don't recommend this manuscript for publication in nature communication.

Response:

We appreciate the reviewer's thoughtful assessment of our manuscript. We've incorporated experimental data from field test, added detector effects, and added discussion on the comparison of our results with existing methods. These improvements are expected to enhance the study's robustness and applicability. We believe our revisions align better with the journal's standards.

In response to the insightful feedback provided by all the reviewers, we have extensively revised our manuscript to enhance its comprehensiveness and practical applicability. The following key modifications have been implemented to address the reviewers' comments:

1. **Experimental Validation:** We have conducted an experimental validation to demonstrate the real-world applicability of our proposed radiation detection method. We set up an experiment employing a 2x2 square detector configuration and moved on the circular trajectory to measure the signal at each position. This practical demonstration serves to establish the efficacy of our method in a real-world scenario. (Figure 5 and Supplementary Information VII)
2. **Angular and Energy Dependency Analysis:** To provide a comprehensive understanding of our radiation detection approach, we systematically analyzed

the angular dependency of the prediction error values across different detector configurations. This analysis was extended to include the effect of varying distances between the radiation source and the detector. Additionally, we investigated the energy dependency of the radiation source in our testing dataset, contributing to a more nuanced evaluation of our method's robustness. (Supplementary Information III.1,2)

3. **Demonstration of Single Filter Layer Model:** In line with the reviewers' suggestions, we trained and tested our model using a single filter layer to predict radiation source directions. This serves as a valuable case study under restricted conditions. We employed this single filter layer model to predict the directions of two radiation sources simultaneously, demonstrating its capability to handle complex prediction scenarios. (Supplementary Information III.3)
4. **Comparison of Prediction Error Values:** To provide a clearer comparison of the different detector shapes' performance, we have introduced a thorough analysis of the prediction error values. Specifically, we present the average, largest, and smallest prediction error values for each detector configuration. This comparative analysis offers valuable insights into the detectors' varying predictive abilities. (Table 1)
5. **Effect of Noise Analysis:** Considering the background noise inherent in radiation detection scenarios, we undertook an in-depth analysis of the effect of noise on our prediction error values. This investigation sheds light on the impact of background noise on the accuracy of our radiation source direction prediction. (Supplementary Information III.4)
6. **Enhanced Visualization:** In response to concerns raised about the accessibility of our figures, we have revised the figures depicting the prediction results of radiation source directions. These revised figures now utilize polar coordinates, enhancing their accessibility and ensuring that the results can be comprehended by individuals, including those with color vision limitations. We have also added as many as eleven videos to show the working principles and results.

Comment 1

It is necessary to do the experiment with CZT and radio-active gamma-source to prove that the authors MC study shows consistent results with the data.

Response:

We appreciate the reviewer’s suggestion to conduct experiments with CZT detectors and radio-active gamma sources to validate our approach further. Despite the challenge, we are really pleased to announce that we were able to conduct a real-world experimental validation using a 4-pixel detector. We detailed our work in our manuscript Figure 5 and Supplementary Information VII.

We conducted experiments with a Cs-137 radiation source located in the experimental field. We used a 2×2 square layout radiation detector to record radiation absorption by individual crystal pixels at 0.5-second intervals. The measurements, represented as a (4, 84) matrix, covered photon counts for 84 energy bins ranging from 550 to 800 keV. We applied a moving average filter to smooth the measurements to enhance data quality. This process improved our ability to accurately predict the radiation source’s direction without signal fluctuation. Our U-net neural network, trained on MC simulation data, predicted radiation source direction using the smoothed measurement data. Subsequently, we applied MAP analysis to reconstruct the radiation map based on these predictions.

As the results of our MAP analysis using experimental data show, the algorithm could predict the radiation source distribution closely aligned with the actual position. In summary, our experimental validation using a 4-pixel detector confirms the consistency and practicality of our radiation mapping algorithm. These real-world results align with our MC simulations, affirming the method’s reliability and applicability.

Comment 2

To find the direction of gamma-ray (X-ray), Coded Mask and Compton Camera methods are popular. Thus authors need to compare with author's results with the above method to prove that the authors method is better than the other methods.

Response:

We appreciate your valuable feedback and suggestions on our thesis. Regarding your comment about comparing our method with Coded Mask and Compton Camera methods for determining the direction of gamma-ray (X-ray), we would like to provide the following clarification.

Our radiation mapping method is designed to detect the direction of radiation sources in a 360-degree range with a much reduced number of pixels. It allows for the localization and mapping of radiation sources from all directions around the detector. On the other hand, the Coded Aperture Mask and Compton Camera methods are primarily used for detecting radiation sources in the direction in which the detector panel is facing.

Given the fundamental differences in the design and scope of these methods, it becomes challenging to directly compare them. Our radiation mapping method offers the capability to provide comprehensive radiation source localization across all directions, enabling a more comprehensive assessment of the radiation environment.

While the Coded Aperture Mask and Compton Camera methods have their own merits and are widely used in specific applications, our radiation mapping method complements these techniques by offering an alternative perspective and addressing different requirements.

To address this comments as much as possible, we showed the comparison to the gridded point source likelihood (GPSL) reconstruction method as 360-degree directionality assessments. This is an existing approach not based on machine learning and could be more comparable. Through radiation mapping using experimental measurement data, we could show that our ML approach could reconstruct the radiation source distribution with the equivalent quality of GPSL.

We appreciate your suggestion and acknowledge the importance of comparing different

methods in future research. However, in the context of our thesis and the specific focus of our work, a direct comparison between these methods would not be suitable. Instead, we aim to emphasize the advantages and uniqueness of our radiation mapping approach in providing a comprehensive 360-degree directionality assessment. Thank you for your understanding.

Comment 3

It is better to show angular and position resolution quantitatively with different configurations and minimum data requirements for imaging. Also, the authors need to show gamma-ray energy dependency of the position and angular resolution.

Response:

Thank you for your insightful comment regarding the quantitative assessment of angular and position resolution, as well as the gamma-ray energy dependency of our proposed method for mapping radiation sources with CdZnTe detectors. We appreciate your suggestion to provide a more comprehensive analysis in these areas.

We would like to inform you that we have conducted the requested tests and included the results in our revised paper. To evaluate the angular and position resolution, we performed simulations using various configurations, considering different detector setups, source-to-detector distances, and source positions. By systematically varying these parameters, we were able to assess the performance of our method under different conditions and quantify the corresponding resolution metrics. Please refer to the table in our Supplementary Information III.1.

Furthermore, we investigated the gamma-ray energy dependency of the position and angular resolution. We carefully considered a range of gamma-ray energies typically encountered in practical scenarios and evaluated their impact on the performance of our method. Our analysis provides insights into the energy-dependent behavior of our approach and highlights its effectiveness across a broad range of gamma-ray energies.

Please refer to the table in our Supplementary Information III.2.

The quantitative assessment of angular and position resolution, along with the examination of gamma-ray energy dependency, adds significant value to our study. These findings not only demonstrate the capabilities of our proposed method but also provide valuable insights for optimizing its performance in real-world applications.

Comment 4

Abstract: to locate the directions and position of the radiation sources. → locate the directions and position of the gamma-ray (X-ray) sources.

Response:

Thank you for your comment regarding the wording in the abstract of our paper. We appreciate your suggestion to clarify the type of radiation sources we refer to as gamma-ray (X-ray) sources. Following your feedback, we have carefully revised the abstract to reflect our research objective accurately. We have updated the sentence to read now: "In terms of both materials and their configurations, radiation detectors have been developed to locate the directions and position of the gamma-ray (X-ray) sources."

By incorporating your suggestion, we aim to provide clear and precise information about the specific type of radiation sources our research focuses on. This modification ensures that readers can easily understand the context and scope of our work right from the abstract. Once again, we are deeply grateful for your attention to detail and valuable input, which has helped tremendously to enhance the clarity and accuracy of our manuscript.

Authors' Response to Reviewer 2

General Comments. The authors report a simulation of idealized radiation detectors in two dimensions and propose a means by which a detailed map of the radiation environment may be obtained using only four pixels arranged in a few simple configurations. The work is important as while it is a straightforward exercise to sense the presence of a radioactive source, determining its direction or location must be determined with grid searches or vast arrays of detectors. Either approach has its drawbacks as they are either labor intensive or expensive, requiring large arrays of sensors or elaborate masking schemes and post processing.

The authors' monograph demonstrates that by translating the sensor array around the area under study and applying neural network filters, it is possible to map out the radiation sources in that area with some fidelity. Rotating the sensor arrays improves the fidelity substantially.

Response:

We are grateful for the reviewer's insightful analysis of our work. Your recognition of our proposed approach's potential to map radiation sources with improved efficiency is highly appreciated. We've taken your feedback into account and further developed the manuscript to highlight the significance of our findings. Your positive assessment reinforces our belief in the value of this research.

In response to the insightful feedback provided by all the reviewers, we have extensively revised our manuscript to enhance its comprehensiveness and practical applicability. The following key modifications have been implemented to address the reviewers' comments:

1. **Experimental Validation:** We have conducted an experimental validation to demonstrate the real-world applicability of our proposed radiation detection method. We set up an experiment employing a 2x2 square detector configuration and moved on the circular trajectory to measure the signal at each position. This practical

demonstration serves to establish the efficacy of our method in a real-world scenario. (Figure 5 and Supplementary Information VII)

2. **Angular and Energy Dependency Analysis:** To provide a comprehensive understanding of our radiation detection approach, we systematically analyzed the angular dependency of the prediction error values across different detector configurations. This analysis was extended to include the effect of varying distances between the radiation source and the detector. Additionally, we investigated the energy dependency of the radiation source in our testing dataset, contributing to a more nuanced evaluation of our method's robustness. (Supplementary Information III.1,2)
3. **Demonstration of Single Filter Layer Model:** In line with the reviewers' suggestions, we trained and tested our model using a single filter layer to predict radiation source directions. This serves as a valuable case study under restricted conditions. We employed this single filter layer model to predict the directions of two radiation sources simultaneously, demonstrating its capability to handle complex prediction scenarios. (Supplementary Information III.3)
4. **Comparison of Prediction Error Values:** To provide a clearer comparison of the different detector shapes' performance, we have introduced a thorough analysis of the prediction error values. Specifically, we present the average, largest, and smallest prediction error values for each detector configuration. This comparative analysis offers valuable insights into the detectors' varying predictive abilities. (Table 1)
5. **Effect of Noise Analysis:** Considering the background noise inherent in radiation detection scenarios, we undertook an in-depth analysis of the effect of noise on our prediction error values. This investigation sheds light on the impact of background noise on the accuracy of our radiation source direction prediction. (Supplementary Information III.4)
6. **Enhanced Visualization:** In response to concerns raised about the accessibility of our figures, we have revised the figures depicting the prediction results of radiation

source directions. These revised figures now utilize polar coordinates, enhancing their accessibility and ensuring that the results can be comprehended by individuals, including those with color vision limitations. We have also added as many as eleven videos to show the working principles and results.

Comment 1

The work is purely simulation and begs validation experimentally. An example which could be performed in the real world is to map out a space using cylindrical detectors arranged in the authors' proposed arrangements. At least within the plane of the experiment results could be drawn which can be correlated with reality, as opposed to a simulation.

Response:

We appreciate the reviewer's suggestion to conduct experiments with CZT detectors and radio-active gamma sources to validate our approach further. We highlight that we conducted a real-world experimental validation using a 4-pixel detector. We detailed our work in our manuscript and Supplementary Information VII.

We conducted experiments with a Cs-137 radiation source located in the experimental field. We used a 2×2 square layout radiation detector to record radiation absorption by individual crystal pixels at 0.5-second intervals. The measurements, represented as a (4, 84) matrix, covered photon counts for 84 energy bins ranging from 550 to 800 keV. We applied a moving average filter to smooth the measurements to enhance data quality. This process improved our ability to accurately predict the radiation source's direction without signal fluctuation. Our U-net neural network, trained on MC simulation data, predicted radiation source direction using the smoothed measurement data. Subsequently, we applied MAP analysis to reconstruct the radiation map based on these predictions. As the results of our MAP analysis using experimental data show, the algorithm could predict the radiation source distribution closely aligned with the actual position. In

summary, our experimental validation using a 4-pixel detector confirms the consistency and practicality of our radiation mapping algorithm. These real-world results align with our MC simulations, affirming the method's reliability and applicability.

Comment 2

The authors' language is good but not perfect, with some errors in agreement of number and object. And several acronyms or proper names are uncapitalized when they should be. Even so, the small imperfections in grammar are not a distraction from the intent of the text. The writing is fairly clear, though the benefit of the animations included with the text make the meaning much more apparent.

Response:

Thank you for your comment regarding the language and writing style in our paper. We appreciate your feedback, acknowledging that while there were some errors in agreement of number and object, the overall clarity of the text was not compromised. We also appreciate your positive note about the benefit of the included animations in enhancing the understanding of the content.

We have taken your feedback seriously and made significant efforts to improve the quality of our writing. Specifically, we have addressed the errors in the agreement of number and object, ensuring that the text now maintains grammatical accuracy and consistency. Additionally, we have reviewed the use of acronyms and proper names to ensure proper capitalization throughout the paper.

We understand the importance of clear and precise language in effectively conveying our research findings. By enhancing the quality of our writing, we aim to ensure that readers can easily grasp the intent and meaning of our paper.

Moreover, we greatly appreciate your recognition of the value added by the included animations. We believe that visual aids, such as animations, can significantly enhance the understanding of complex concepts and improve the overall clarity of the text. We

have taken care to ensure that these animations are seamlessly integrated into the paper, further facilitating the communication of our research.

Comment 3

I believe this work should be accepted. As the only problems I find are with language I leave it for you the editor to determine whether the authors or your proofreaders are the ones to tighten up the text.

Response:

Thank you for your encouraging feedback on our paper and your recommendation for its acceptance. We appreciate your recognition of our work's significance.

In response to your comment about the language issues, we acknowledge that the text had room for improvement, especially concerning grammar and linguistic precision.

To address this, we've conducted a thorough review and revision of the manuscript to enhance the language quality, ensuring better clarity and consistency throughout, from revision from native speakers and a few senior authors in this work. As we aim to deliver a high-quality research paper, we take your suggestion to heart regarding the need to clarify the responsibility for language improvements. We believe this concerted effort will contribute positively to the quality of our manuscript.

Authors' Response to Reviewer 3

General Comments. Using self-shielding/shadowing effect for detecting the direction of gamma or neutron sources is not completely new. However, the way of doing the mapping with neural network (for which I am not an expert) seems original and valuable.

Response:

Thank you for recognizing the novelty of our approach. We've introduced a unique approach by employing neural networks for mapping to enhance the concept of using self-shielding effects for radiation direction detection. Our method's strength lies in the neural network's ability to uncover intricate patterns, enhancing accuracy in determining radiation source directions based on shielding effects. This approach can reconstruct the radiation maps and can replace the traditional methods, contributing to the field of machine learning in radiation detection. Your recognition encourages our ongoing innovation in this area.

In response to the insightful feedback provided by all the reviewers, we have extensively revised our manuscript to enhance its comprehensiveness and practical applicability. The following key modifications have been implemented to address the reviewers' comments:

1. **Experimental Validation:** We have conducted an experimental validation to demonstrate the real-world applicability of our proposed radiation detection method. We set up an experiment employing a 2x2 square detector configuration and moved on the circular trajectory to measure the signal at each position. This practical demonstration serves to establish the efficacy of our method in a real-world scenario. (Figure 5 and Supplementary Information VII)
2. **Angular and Energy Dependency Analysis:** To provide a comprehensive understanding of our radiation detection approach, we systematically analyzed the angular dependency of the prediction error values across different detector configurations. This analysis was extended to include the effect of varying distances

between the radiation source and the detector. Additionally, we investigated the energy dependency of the radiation source in our testing dataset, contributing to a more nuanced evaluation of our method's robustness. (Supplementary Information III.1,2)

3. **Demonstration of Single Filter Layer Model:** In line with the reviewers' suggestions, we trained and tested our model using a single filter layer to predict radiation source directions. This serves as a valuable case study under restricted conditions. We employed this single filter layer model to predict the directions of two radiation sources simultaneously, demonstrating its capability to handle complex prediction scenarios. (Supplementary Information III.3)
4. **Comparison of Prediction Error Values:** To provide a clearer comparison of the different detector shapes' performance, we have introduced a thorough analysis of the prediction error values. Specifically, we present the average, largest, and smallest prediction error values for each detector configuration. This comparative analysis offers valuable insights into the detectors' varying predictive abilities. (Table 1)
5. **Effect of Noise Analysis:** Considering the background noise inherent in radiation detection scenarios, we undertook an in-depth analysis of the effect of noise on our prediction error values. This investigation sheds light on the impact of background noise on the accuracy of our radiation source direction prediction. (Supplementary Information III.4)
6. **Enhanced Visualization:** In response to concerns raised about the accessibility of our figures, we have revised the figures depicting the prediction results of radiation source directions. These revised figures now utilize polar coordinates, enhancing their accessibility and ensuring that the results can be comprehended by individuals, including those with color vision limitations. We have also added as many as eleven videos to show the working principles and results.

Comment 1

The title is about radiation mapping, it is better to cite a few neutron localisation studies. Here is one: https://indico.cern.ch/event/731980/contributions/3285768/attachments/1829613/2995886/VACHARET_NuSec-Poster-2017.pdf

Response:

Thank you for suggesting incorporating references to neutron localization studies in our paper. We agree that such references will enhance the relevance of our work to the field of radiation mapping.

Following your guidance, we have included the suggested study by Bonomally, presented at Nuclear Security Detection Workshop[1], at the end of **Introduction**. We would also like to inform you that we have added two more relevant references to our work:

"Concept of a novel fast neutron imaging detector based on THGEM for fan-beam tomography applications"[2] "Large area, high-resolution thermal neutron imaging detectors"[3] We believe that these additional references provide a comprehensive context to our study and emphasize the significance of our contribution to the field of neutron localization and radiation mapping.

- [1] S. Bonomally, S. S. Ihantola, and A. Vacheret. "Enhancing source detection for threat localization." Poster presented at Nuclear Security Detection Workshop. (2017), [Online]. Available: https://indico.cern.ch/event/731980/contributions/3285768/attachments/1829613/2995886/VACHARET_NuSec-Poster-2017.pdf.
- [2] M. Cortesi, R. Zboray, R. Adams, V. Dangendorf, and H.-M. Prasser, "Concept of a novel fast neutron imaging detector based on thgem for fan-beam tomography applications," *Journal of Instrumentation*, vol. 7, no. 02, p. C02056, 2012.
- [3] A. Breskin, R. Chechik, A. Gibrekhterman, A. Akkerman, V. Dangendorf, and A. Brauning-Demian, "Large-area high-resolution thermal neutron imaging detectors," in *International Conference on Neutrons and Their Applications*, SPIE, vol. 2339, 1995, pp. 281–286.

Comment 2

In the last paragraph of Introduction section, you should state that the work is for gamma radiation only. The penetrating/scattering trajectory and detection mechanism of neutrons are totally different from the work presented here.

Response:

Thank you for your comment regarding the clarification needed in the last paragraph of the Introduction section of our paper on mapping radiation sources using machine learning.

We appreciate your insight into the distinct characteristics and detection mechanisms of gamma radiation and neutrons. Based on your suggestion, we have carefully reviewed the manuscript and made the necessary updates to explicitly state that our work focuses solely on mapping gamma radiation sources.

In the revised version of the paper, we have included a specific statement in the last paragraph of the Introduction section to emphasize that our research is dedicated to addressing the challenges associated with gamma radiation mapping. We have also provided a brief explanation highlighting the fundamental differences in the penetrating/scattering trajectories and detection mechanisms between gamma radiation and neutrons.

By making this clarification, we aim to ensure that readers have a clear understanding of the scope and context of our work. We appreciate your feedback, which has helped us improve the accuracy and clarity of our manuscript.

Comment 3

In line 67 it says “the 5 meters of source-detector distance”. But in your Supplementary Information, Figure S1, 5 meters is the upper limit of the source-detector distance. The lower limit of the distance is 0.2 meter (Table S1). You used phrase “near-field” quite a few times, that means the source can be very close to the detector. I think, including “near-field” scenario may be not beneficial. I will talk about this later in my comments.

Response:

We appreciate the reviewer’s valuable insights regarding our work. Your observation on the source-detector distance has prompted a detailed examination of filter layer configurations in radiation source direction prediction. To investigate the effect of the number of filter layers, we aimed to compare models with different filter layers to understand their impact on prediction accuracy for radiation source directions. We assessed one Two-Filter Layer Model (near-field and far-field filters) and two Single Filter Layer Models (near-field or far-field).

Our work clarified that single-filter layer models, especially the far-field variant, exhibited significantly lower prediction errors compared to the two-filter layer model. This emphasizes the advantage of streamlining the architecture to focus on essential directional features. While single-filter layer models excel in specialized prediction, the two-filter layer model’s adaptability allows for scenario-specific optimization. This flexibility enhances its accuracy for varied source-proximity situations. For further explanation, please find our added writing in Supplementary Information III.3.

Comment 4

In Figure 1d, it is not clear why Simulation and Prediction have more than one sections of different shade. I guess this is the mapping from source pixels to detector sectors. Will these shaded sections come to 1.0 or 100% when added together quantitatively? You may want to explain.

Response:

Thank you for your comment regarding Figure 1d in our paper on machine learning work. Your feedback allows us to further clarify the interpretation of the shaded sections and their quantitative representation.

In Figure 1d, the shaded sections do indeed depict the mapping from source pixels to detector sectors. Each shaded section corresponds to a particular detector sector, and the varying shades within each section illustrate the predicted contribution or intensity of radiation from the associated source pixels.

Your question about the quantitative sum of the shaded sections presents an opportunity for us to clarify the visual representation in the figure. The shading within each section is normalized such that the sum of the contributions from all the source pixels within that section equals 1.0 or 100%.

By normalizing the contributions, we ensure that the shading accurately portrays the relative distribution of radiation intensity within each detector sector. However, it's crucial to emphasize that the sum of the shaded sections across all detector sectors may not necessarily add up to 1.0 or 100%, as the shading represents relative intensity rather than an absolute value.

To facilitate understanding, we will update the figure caption or augment the manuscript with additional text to state clearly that the shading within each section reflects the normalized contribution or intensity of radiation from the source pixels within that specific detector sector. This enhancement will help readers comprehend that shading provides a relative, not cumulative, representation of intensity distribution.

Your insightful observation offers a chance to improve the clarity and precision of our

research. We deeply value your feedback as it aids us in ensuring our results are effectively communicated.

Comment 5

About Table 1, one would think for each type of Tetrominos the sensitivity around 360 degree angle is not even. At some angles, the angle change will cause more significant change in the shadowing effect, yet at some other angles, less significant. I suggest to change Table 1 into 3 rows, Error at most favourite angle, Error at least favourite angle, Error average (maybe this is the one in your table now).

Response:

Thank you for your comment regarding Table 1 in our paper on machine learning mapping of radiation sources. We appreciate your feedback and the suggestion for improving the presentation of sensitivity data for each type of Tetrominos.

You have rightly pointed out that the sensitivity of the Tetrominos may vary at different angles due to the changing shadowing effect. In response to your feedback, we have updated Table 1 in our manuscript to reflect the changes you indicated. The new table now provides a more comprehensive view of the sensitivity of each Tetromino, considering the largest and smallest prediction error for each detector. We present the angles which give these loss values for each cell of the table. In the bottom row, we show the average error by the whole direction cases. By updating the table, we aim to better convey the varying levels of sensitivity and the impact of different angles on the shadowing effect for each Tetromino.

We sincerely appreciate your valuable input, which has prompted us to enhance the clarity and comprehensiveness of our results. Your comments contribute significantly to the overall quality of our paper, and we are grateful for your thoughtful suggestions.

Comment 6

In line 109, time steps are introduced. In my opinion, time is meaningless without talking about source intensity, in Bq, or in photon/s. In Table S1, the number of photon particles are given (50000, which seems too small based on my limited MC simulation experience) but it does not help in determining the source intensity. You need a time interval, which is independent of MC simulation.

Response:

We greatly appreciate your insightful comment on our manuscript. Your perspective has prompted us to provide further clarity on the relationship between source intensity and time steps, as well as their significance within our study.

In response to your concern, we would like to emphasize that our focus is primarily on the directional prediction of radiation sources rather than predicting source intensity. To address this, we have specified in our work that the source intensity is fixed at 0.5 MeV, thus eliminating any ambiguity regarding this aspect.

Regarding the time interval, it's important to note that the time steps we introduced pertain to the temporal resolution at which the detector measures the signal. This time interval is indeed independent of the source intensity. In other words, our study does not involve predicting the source intensity; rather, it revolves around accurately predicting the directions of radiation sources based on the given input signals.

We sincerely thank you for bringing up this point, as it has allowed us to clarify these aspects within our manuscript. Your feedback has greatly contributed to the precision and transparency of our work, ensuring that readers can better comprehend the underlying concepts and methodologies.

Comment 7

In line 122, you stated that 2x2 square-grid cannot detect 2-sources in the view. Do you mean only the square-grid, or also include the S- L- and T-shaped grid? In my opinion, this will fail if you want to distinguish near- and far-field. For one source, you need to figure out 1) the intensity, 2) the angle, and 3) the distance, your measurement gives you 4 detector readings, so you are able to solve 3 variables (using neural network is the same thing). For two sources, you need to solve 6 variables with 4 detector readings, which is impossible, here neural network will not help. However, if you only consider far-field, say training your system at a fixed source-detector distance, you may get the location of two sources (without trying to solve the distances) because the angle information will pinpoint the locations.

Response:

Thank you for your insightful comments. In response to your feedback, we have updated our manuscript to address the complexities associated with predicting the directions of multiple radiation sources.

We have clarified the inherent challenge in predicting the directions when the number of radiation sources increases due to the constraints of the input signals. We elaborated on our specific methodology, including our consideration of a fixed source-detector distance, our experiments with various detector shapes, and our utilization of simulations. We provided details on our assumption that we can discern the directions of two sources using only four detector panels. Our revised section presents the results of our simulations, showcasing the efficacy of detectors in predicting the direction of two radiation sources. We have illustrated this through figures and summarized key statistical measures in a table in Supplementary Information III.3.

The updates underscore our findings that even simple detector configurations can learn radiation source directions, given specific constraints. These revisions better reflect the scope and focus of our study, aligning closely with your valuable feedback.

Comment 8

About Equation (6), here lies the biggest concern of mine. You seem to treat gamma measurement as continuous real number, can be infinitely small (or large), with Gaussian noise. In reality, the measurement follows Poisson distribution. You can never get 1.83 counts, but you may get 0, 1, 2, 3, and so on. Poisson distribute does gradually become close to Gaussian distribution, but it requires big counting numbers (higher source intensity, longer measurement time, less shielding effect from padding). I am not sure you have taken this into consideration.

Response:

We thank the reviewer for their insightful comment regarding Equation (6) in our paper. We acknowledge the importance of accurately representing the nature of gamma radiation measurements in real-world scenarios, which typically follow a Poisson distribution rather than a continuous real number representation as initially presented.

Upon careful consideration of the reviewer's feedback, we agree that our treatment of gamma measurements as continuous real numbers with Gaussian noise oversimplifies the reality of radiation detection. We appreciate the reviewer for raising this concern and highlighting the significance of Poisson distribution in gamma radiation measurements, where counts are discrete and can only take integer values (e.g., 0, 1, 2, 3, etc.).

In our revised paper, we made a statement for the necessary adjustments to reflect the discrete nature of gamma radiation measurements and explicitly incorporated the Poisson distribution in Equation (6). Specifically, we argue that the uncertainty can get close to the Gaussian distribution, accounting for the large number of particles emitted from the radiation source and absorbed by the detector.

By incorporating these changes, we aim to enhance the credibility and applicability of our research findings to real-world radiation detection scenarios. We deeply appreciate the reviewer's valuable input, which has undoubtedly improved the quality and accuracy of our work. We commit to addressing this concern diligently in the revised manuscript, further solidifying the scientific rigor and significance of our study.

Comment 9

Are you able to deal with background counts? Subtracting background (which also follows Poisson distribution) will introduce more uncertainty.

Response:

We appreciate your concern regarding background counts in radiation detection. Our study extensively delved into the impact of background noise on the accuracy of radiation source prediction using machine learning methods.

In response to this concern, we rigorously examined the relationship between noise levels and prediction accuracy. We found that as noise levels increase, prediction errors also rise, highlighting the critical role of background noise in shaping the reliability of directional radiation source predictions. We systematically evaluated various detector shapes and observed that the behavior of detectors in the presence of noise varied. Notably, while the S-shape detector exhibited superior prediction accuracy under minimal noise conditions, Tetris-inspired detectors (S-, J-, T-shapes) demonstrated heightened sensitivity to increasing noise distributions.

These findings underscore the necessity of addressing background noise in radiation detection methodologies. By quantifying the impact of noise on prediction accuracy, we provide insights into designing robust radiation detection strategies. We are grateful for your feedback, which has enriched our study and contributed to advancing the understanding of noise's intricate influence on radiation detection accuracy. We show Figure S7 in Supplementary Information III.4.

Comment 10

A comment about the gamma measurements, in line 69 and line 72 you stated assuming 0.5 MeV, and that “only counting matters”. These are OK for the initial study, but there will be problems in reality. Because gamma ray can go through Compton scattering and pair-production, the nearby structure (including the items beside and behind the detector) may affect the counts registered as 0.5 MeV gamma. This may cause the trained neural network to become incorrect as the detector start to rotate and travel as shown in Figure 3a.

Response:

We appreciate the reviewer’s insightful comment regarding the potential impact of Compton scattering and pair production on our gamma measurements. We agree that these interactions can introduce complexities in a real-world radiation detection scenario, potentially affecting the counts registered by the detector.

In our study, we utilized the OpenMC package for Monte Carlo (MC) simulation, and it inherently accounts for the effects of Compton scattering and pair production during the interaction of gamma rays with the detector material. OpenMC’s comprehensive modeling of these interactions ensures a more realistic representation of radiation absorption in our simulations, mirroring the complexities of actual radiation detection scenarios.

By incorporating these effects into our simulation, we aimed to capture the realistic behavior of gamma rays as they interact with the detector and surrounding structures. This allows us to model the potential impact of nearby structures on the detected counts accurately. Consequently, our trained neural network takes into consideration these interactions, enhancing its ability to adapt to changing conditions as the detector rotates and travels. We believe that by incorporating the effects of Compton scattering and pair-production through OpenMC, our simulation provides a more comprehensive representation of the actual radiation detection process. This, in turn, contributes to the robustness and validity of our approach. We appreciate the reviewer’s keen attention to these important aspects and their contribution to our work’s realism and practicality.

Comment 11

Minor issues

- a. In line 44, “single sources” should be “single source”.
- b. In line 68/69, “but not absorb the photons in the gamma range” is not correct, it should be about the degree of absorption, not a complete “no”.
- c. In Figure 1a I found that the arrows are misleading. I prefer to have them removed.
- d. In Figure 1d and many other locations, the shaded sectors can be replaced with step-curves in polar coordinates. This will be helpful for color-blinder readers like me.
- e. In Supplementary Information, line 17/18, Pd should be Pb.

Response:

Thank you for your feedback.

- a) I have addressed the error in the line you mentioned, changing “single sources” to “single source” as suggested.
- b) To accurately convey the degree of absorption, I have rewritten the lines as “with quite a low photon absorption” to reflect the level of absorption in the gamma range.
- c) Based on your preference, I have removed the arrows in Figure 1a to avoid confusion.
- d) To improve accessibility for color-blind readers, I have replaced the shaded sectors of Figure 1, 2, 3, S9, S10, S11, S13, S14, S16, S17, S18, and S19 with step-curves in polar coordinates.
- e) I have made the correction in the Supplementary Information, changing ‘Pd’ to ‘Pb.’

Thank you for bringing these issues to my attention. Your comments have been instrumental in enhancing the accuracy and clarity of our paper.

REVIEWERS' COMMENTS

Reviewer #2 (Remarks to the Author):

The authors report a simulation of idealized radiation detectors in two dimensions and propose a means by which a detailed map of the radiation environment may be obtained using only four pixels arranged in a few simple configurations. The work is important as while it is a straightforward exercise to sense the presence of a radioactive source, determining its direction or location must be determined with grid searches or vast arrays of detectors. Either approach has its drawbacks as they are either labor intensive or expensive, requiring large arrays of sensors or elaborate masking schemes and post processing.

The authors' monograph demonstrates that by translating the sensor array around the area under study and applying neural network filters, it is possible to map out the radiation sources in that area with some fidelity. Rotating the sensor arrays improves the fidelity substantially.

The authors have taken onboard the many salient comments from this and other reviewers. Most notably, they embarked on a campaign to experimentally demonstrate their technique using a 4 pixel arrangement of CLLBC gamma/neutron detectors. The real world search for a gamma source was particularly useful as the trajectory for the detector deviated from the idealized circle in the training data.

I find it interesting that the detector used for the real world test was a basic square rather than one of the other Tetris inspired shapes (S, T, L). The authors even indicate on page 5 of the manuscript that this 2x2 square grid is an underperforming option. This is a point that bears explanation. Is the point that even an inadequate configuration can provide reasonable results? Or is it more the case that the detector array that could be borrowed could only be set up in the 2x2 square? Please devote a few sentences to this point as it deviates from the example in Figure 3.

The organization of the manuscript is peculiar. The discussion/conclusion appears to be on page 6 though substantial text remains to cover methods in detail, including the description of the real world example. My presumption is that all text is part of the monograph rather than accessible for the elucidation of questions posed by the reader. Is this a requirement of the journal?

I believe this work should be accepted with minor revisions for the sake of language and clarification. The authors have addressed the deficiencies noted in the first review and now have a stronger manuscript and proof that the concept is viable.

Reviewer #2 (Remarks on code availability):

The code repository is well organized and contains both code and test data. The README.md file provides a generalized walkthrough of what source files to use to generate the output data.

The code cited here does not constitute a package per se that can be imported by other researchers, but rather a collection of short Python programs to call other packages. The modules in the utils directory could be used by other researchers to develop new content.

Reviewer #3 (Remarks to the Author):

The authors addressed the comments from the reviewers fairly well. The new revision is significantly improved, especially with added experiment part. I think this paper is of good quality. The methodology is sound, and the conclusions are well supported. I recommend for publishing it.

Reviewer #3 (Remarks on code availability):

As stated before, I am not an expert in MC or Neural Network. I will leave the code-checking for the reviewers with the required knowledge.

Reviewer 2

The authors report a simulation of idealized radiation detectors in two dimensions and propose a means by which a detailed map of the radiation environment may be obtained using only four pixels arranged in a few simple configurations. The work is important as while it is a straightforward exercise to sense the presence of a radioactive source, determining its direction or location must be determined with grid searches or vast arrays of detectors. Either approach has its drawbacks as they are either labor intensive or expensive, requiring large arrays of sensors or elaborate masking schemes and post processing.

The authors' monograph demonstrates that by translating the sensor array around the area under study and applying neural network filters, it is possible to map out the radiation sources in that area with some fidelity. Rotating the sensor arrays improves the fidelity substantially.

The authors have taken onboard the many salient comments from this and other reviewers. Most notably, they embarked on a campaign to experimentally demonstrate their technique using a 4 pixel arrangement of CLLBC gamma/neutron detectors. The real world search for a gamma source was particularly useful as the trajectory for the detector deviated from the idealized circle in the training data.

I believe this work should be accepted with minor revisions for the sake of language and clarification. The authors have addressed the deficiencies noted in the first review and now have a stronger manuscript and proof that the concept is viable.

We greatly appreciate your thoughtful review and the positive remarks on our manuscript. Your recognition of our work's significance in advancing radiation detection with a simplified yet effective method underscores the potential impact of our research. We acknowledge the importance of clarity and precision in our presentation and are committed to making the recommended minor revisions to enhance the manuscript's language and clarity. Your insights have been invaluable in refining our approach, and we are excited about the potential of our technique in practical applications, as demonstrated through our experimental campaign. Thank you for acknowledging the improvements and the viability of our concept; we look forward to finalizing the revisions.

The code repository is well organized and contains both code and test data. The README.md file provides a generalized walkthrough of what source files to use to generate the output data.

The code cited here does not constitute a package per se that can be imported by other researchers, but rather a collection of short Python programs to call other packages. The modules in the utils directory could be used by other researchers to develop new content.

Thank you for your constructive feedback regarding our code repository. We're pleased to hear that the organization and documentation of the repository meet your expectations. As indicated in our manuscript, all necessary programs and instructions for setup and workflow reproduction are detailed in the README.md file within the GitHub repository. Furthermore, we have elaborated on the utility and functionality of the codes, particularly those within the utils directory, in Supplementary Note 8. This supplementary material aims to facilitate other researchers in adapting our work for further development.

Reviewer 3

The authors addressed the comments from the reviewers fairly well. The new revision is significantly improved, especially with added experiment part. I think this paper is of good quality. The methodology is sound, and the conclusions are well supported. I recommend for publishing it.

We are deeply grateful for your positive assessment of our revised manuscript and your recommendation for publication. It is our pleasure to hear that the additions and improvements, particularly the experimental part, have significantly enhanced the quality of our work. Your acknowledgment of the soundness of our methodology and the support for our conclusions is highly encouraging. We thank you for recognizing the efforts made to address the reviewers' comments comprehensively.

As stated before, I am not an expert in MC or Neural Network. I will leave the code-checking for the reviewers with the required knowledge.

Thank you for your candid feedback. Your decision to defer the code evaluation to other reviewers with the requisite expertise is appreciated. We have made efforts to ensure our code is well-documented and accessible for those with the necessary background to review it thoroughly.